



# Technical note: A novel technique to improve the hydrological estimates at ungauged basins by swapping workspaces

Muhammad Uzair Qamar[1], Muhammad Azmat[2], Muhammad Usman[1,3], Daniele Ganora[4], Muhammad Adnan Shahid[5], Faisal Baig[6], Sumra Mushtaq[7]

*Correspondence to*: Muhamad Usman (muhammad.usman@uni-wuerzburg.de)

[1]Department of Irrigation and Drainage, Faculty of Agricultural Engineering and Technology.

[2]Institute of Geographical Information Systems (IGIS), School of Civil & Environmental Engineering (SCEE), National University of Sciences and Technology (NUST), IGIS Building (2nd Floor), 44000 Islamabad, Pakistan.

[3]Department of Remote Sensing, Institute for Geography and Geology, Julius Maximillian's University Wuerzburg, Oswald Külpe Weg 86, 97074 Wuerzburg, Germany

[4]Dipartimento di Idraulica, Trasporti ed Infrastrutture Civili, Politecnico di TorinoTurin, Italy

[5]Water Management Research Centre, University of Agriculture, 38040 Faisalabad, Pakistan

[6]Department of Agricultural Engineering, Bahauddin Zakariya University, 60800 Multan, Pakistan

[7]Department of Catchment Hydrology, Helmholtz Centre for Environmental Research UFZ

**Abstract.** The dissimilarity-based methods to perform prediction of flow regimes in ungauged basins have become quite popular in the recent times. Generally, these methods use geomorphological and climatic characteristics of the basins to translate their hydrological properties. However, the methods have been criticized for using selective basin characteristics for the prediction of hydrological data of the basins in the entire study area. Incase these selected descriptors are not strongly related to the hydrological properties of the considered basin; as opposed to the general perception, a considerable magnitude of localized error may be introduced in the final results. To address these drawbacks, we propose a novel technique which assists in identifying a better individual regional model for the prediction of hydrological data at each ungauged basin. The new procedure treats each flow regime as a complete hydrological object. Whereas, the variability in regime shape is determined by using dissimilarity values arranged in a distance matrix executed by considering normalized values of three types of dissimilarities viz; point-to-point dissimilarity, vertical dissimilarity and lateral dissimilarity. On the basis of defined statistical routines, the flow distance matrix is linked with the distance matrices of basin characteristics, acquired by simple comparison of descriptors values, to select most suitable descriptors from the pool of 74 descriptors to form regionalized models. The dissimilarity-based regionalization model thus obtained is primarily coupled with nearest neighbor algorithm to constitute a model space for the initial predictions of the monthly flow regimes. Afterwards, based on the orientation of nearest neighbors of ungauged basin in descriptor space __ the prediction is improved by swapping the model space with the other available models provided the set criteria are fulfilled. The proposed study is conducted in northwestern Italy and the proposed method is tested on the dataset of 124 basins. The basins where the set criteria of model swapping are complied with; the results obtained are statistically better than the initial estimates.





## 1 Introduction

The prediction of flow regimes in general is important for flood mitigation, hydropower generation, dam storage management and irrigation water management. The topic has been widely studied over the last two decades and a number of methods have been proposed for the prediction of hydrological data (Blöschl et al., 2013; Viglione et al., 2013; Qamar et al., 2015, 2016). Among the available methods, dissimilarity-based methods have extensively been used in the recent times owing to their better predictability and simplicity in application (Ganora et al., 2009; Qamar et al., 2016). Theoretically, these methods define hydrological properties of the basins as the function of their climatic, geomorphological and land-use dynamics (also known as descriptors). The descriptors are arranged in a multi-dimensional space to form a workspace in which prediction on hydrologic data is made. The ability of model prediction is generally defined for the selected study-area (or cluster) containing variable number of basins having homogeneous descriptive properties. With the availability of GIS procedure, several descriptors can be computed to investigate the complex basin dynamics: however, the process of model constitution results in a large number of models having almost similar global performances (models exhibiting a very small difference in performance parameters). Afterwards, the predictive model with better global performance is selected from the rest of constituted models by making restrictive assumptions. However, the model selection criteria are not strictly defined but merely the tradeoffs between various statistical parameters (Hall, 2001). Moreover, the selection of the predictive model is based on the redundant information provided by the average predictive performance (of the model) over the selected study area instead for the localized ungauged basin ($u_g$). Therefore, the predictive model, selected from a very competitive domain of models having almost similar predictive abilities, can have the largest prediction uncertainty for the $u_g$ in the study area. Conclusively, it is pertinent for the sake of predictive efficiency to devise such a mechanism that could, somehow, hunch the better model for the considered $u_g$ from the competing models.

We argue that instead of using single model for the overall workspace, there should be a mechanism to define basin-specific model which could statistically execute better predictive results for $u_g$. For this to be done, in our work, we plan to merge the distance based approach with nearest neighbor ($NN$) method to make initial estimates on hydrological data of $u_g$. The estimates will then be improved by swapping the originally selected model with another model, provided the predefined conditions are satisfied.

Unlike other hydrologic entities (e.g., flow duration curve), where flow values are deliberately arranged in the specific order of magnitude; the flow regimes are complex in shape owing to the dependence of flow values on the time parameter. Therefore, the prediction of flow regimes requires not only the predicted flow values to be closer to the actual values but the pattern of occurrence (with respect to time) should also be similar to the actual regime. To reflect this generic difference between flow duration curves and flow regimes in the process of predictive model selection, we used three modes of dissimilarities__ normalized to comprehensively define the dissimilarity between the flow regimes. The hydrological dissimilarities thus executed are related to descriptive dissimilarities, both arranged in the form of distance matrices, to select a so-called original model ($OM$), for the initial estimates. The initial estimates are then potentially improved by swapping the $OM$ with another model having almost similar global performance; defined by $R^2_{adj}$ values and average error generated by the model in the selected workspace ($\Delta$). The



statistical results of swapped model ($SM$) are accepted or rejected by scrutinizing: 1) the extent to which the space
around the $u_g$ is covered ($C_f$) by its $NNs$; and 2) the error generated by $SM$ ($\Delta_{NN}^{SM}$) in predicting the hydrological
data of $NNs$ of $u_g$. We hypothesize that the results of $SM$ can be considered as favorable if and only if $\Delta_{NN}^{SM} < \Delta_{NN}^{OM}$
and $C_f^{SM} > C_f^{OM}$.
**2 Study Area**
The technique in tested in the Northwestern part of Italy. The dataset representing the hydrological and descriptive
characteristics of 124 basins are used in this study (see Figure 1).

86                                          **Figure 1**

The time span of hydrological data varies from 5 years to 52 years with the mean length of 12 years. The runoff data
is extracted from previous publications of former Italian Hydrographic Service updated with the recent
measurements provided by the Regional Environmental Agency (ARPA) of the Piemonte Region. The flow data is
normalized by using global average monthly runoff values at each station. The entire hydrological data is summed
up in Ganora et al. (2013).
The hydrological data is further complimented with the comprehensive compilation of geomorphological and
climatic descriptors obtained for all the selected basins of the study area (Gallo et al., 2013; Farr et al., 2007). The
maximum, minimum and average values of some of the descriptors (out of 74 descriptors) used in our research work
are depicted in Table 1;

96                                          **Table (1)**

The annual flow regimes are executed by summing daily data ($D$) for each month ($M$) to extract an average
monthly representative value through $M_i = \left| \frac{\sum_{j=1}^{N} D_j}{N} \right|_{i=1}^{12}$, where $i$ is the index of the month under consideration, $j$
represents the particular day of the month, and $N$ is the number of days in the month. The monthly runoff regime at
any station is ultimately computed by averaging yearly regimes thus obtaining a single representative flow regime
for each station. The representative regime interprets within-year streamflow variability. This pre-processing forms a
normalized set of data to allow an easier comparison of the flow regimes within the given framework of
dissimilarity. In this work, our primary focus is on the accurate prediction of average monthly runoff magnitudes
and yearly peak flow with respect to time. We are, therefore, interested in a model that is not only able to predict the
correct annual flow volume but also the peak pattern.
**3 Dissimilarity between Regimes**
The dissimilarity between flow regimes is executed by calculating three types of dissimilarities, viz; point to point
distance ($D_{PtP}$), lateral separation ($L_{sp}$), and vertical distance ($V_{sp}$) __ which comprehensively define the difference
in hydrological behavior of the compared basins. The figurative elaboration of three dissimilarities is provided
below in Figure (2);





**Figure (2)**

Assuming, $\{q_{1,S}, q_{2,S}, q_{3,S}, \ldots q_{12,S}\}$ and $\{q_{1,R}, q_{2,R}, q_{3,R}, \ldots q_{12,R}\}$ to be the hydrological data belonging to two gauged

basins $S$ and $R$, respectively; the point to point distance between monthly values can be executed by the following

formula

$D_{PtP} = \sum_{i=1}^{12} |q_{i,S} - q_{i,R}|,$        (1)

where $i$ is the index for monthly values starting from January (when, $i = 1$) and $D_{PtP}$ is the point-to-point difference

between flow regimes of the stations $S$ and $R$. It is important to note that equation (1) is applicable only for

separating flow regimes on the basis of difference in monthly values, but it does not consider the difference in time

between the occurrence of peak flow values (at $S$ and $R$) which is the main characteristic of flow regime (Fig. 2). To

cater the orientation of peak flow in regime, we introduced lateral distance measure ($L_{sp}$) which describes the time

difference between the event of peaks in two regimes by considering initial ($\mu$) and shift ($\sigma$) states of the regimes

using following equation

$L_{sp} = \sum_i |D_{PtP\mu,} - D_{PtP,\sigma}|.$        (2)

The valuation of $L_{sp}$ requires the identification of peaks in the flow regimes that are being compared. In our work,

peaks are considered to be the maximum values in a particular regime. Afterwards a circular procedure is used to

compute lateral separation, in which any of the two regimes is shifted towards the other following least possible

time-steps until both the peaks are exactly underneath each other. For example, in Figure (3) $L_{sp}$ between the flow

regimes belonging to station $S$ and $R$ is calculated. The peak flows of former and later stations occur at 4th and 6th

time-steps, respectively. The shifting of $R$ towards $S$ through 5th and 4th time-steps, takes least number of time-steps

(2-only) to match the peaks; instead of alternative path that requires 10 time-steps (through 7th, 8th, 9th, 10th, 11th,

12th, 1st, 2nd, 3rd, 4th). Each step of peak-shifting is followed by the application of eq. (1), which computes the

dissimilarity between initial and shifted state. It should be noted that the shifted state becomes initial state once the

regime is shifted to the next time step. The dissimilarities obtained during each step are ultimately summed-up to

find the total $L_{sp}$.

**Figure 3**

To ensure that the estimated peaks are not only correct with respect to time but are also closer in terms of

magnitude; a vertical distance measure ($V_{sp}$) which quantifies difference between the peaks is added to the total

distance as

$V_{sp} = |q_{max,S} - q_{max,R}|.$        (3)

Finally, the dissimilarities ($D_{PtP}, L_{sp},$ and $V_{sp}$) are normalized by $\left(\frac{d_i - (d_i)_{min}}{(d_i)_{max} - (d_i)_{min}}\right)$ and added, to calculate a single

representative total dissimilarity value ($D_T$) between the two flow regimes





$$D_T = D_{PtP}^{N_r} + L_{sp}^{N_r} + V_{sp}^{N_r}, \qquad (4)$$
where superscript $N_r$ indicates normalized dissimilarities. A comparison, for $D_T$, is made between 124 stations used
in our work to construct a comprehensive dissimilarity matrix of hydrological data.
Unlike hydrological data, the descriptive data is varying in nature (geomorphological, climatic, etc.). The types of
descriptors used in our work include: (1) single number values (e.g., basin elevation, basin area etc.); (2) monotonic
function, such as hypsographic curve; and (3) complex descriptors like rainfall regimes. The dissimilarity between
the descriptor is computed depending on the type of descriptors. For single value descriptors, absolute difference is
taken between their values. While, in case of monotonic descriptors, eq. (1) is used. Whereas, the dissimilarity
function between regime descriptors is executed in a similar way to that of flow regimes (as $D_T$).
The hydrological and descriptor dissimilarity matrices are expected to assist in the identification of predictive
regional models having efficient temporal and magnitudinal prediction abilities for peak and monthly flow values,
respectively.
**4 Regional Model**
The predictive models are identified by linking descriptor distance matrices with discharge distance matrices
through linear regression to identify the dominating descriptors. The linear model reads as
$$M_H = \beta_0 + \beta_1 (M_D)_1 + \beta_2 (M_D)_2 + \beta_3 (M_D)_3 \dots \beta_i (M_D)_p + \varepsilon, \qquad (5)$$
where $p$ represents the number of descriptors, $\beta_i$ as generic regression coefficient, $\varepsilon$ symbolizes residual element and
$M_D$ depicts descriptor distance matrix transformed into a vector by following a procedure outlined by Lichstein
(2007); which describes, in detail, a methodology for multiple regression (MRM) on distance matrices. The
significance of the regression is quantified through modified Mantel test against 0.05 significance level. The models
sieving through the defined criteria are listed in decreasing $R_{adj}^2$ order, determined by
$$R_{adj}^2 = 1 - (1 - R^2)\frac{n-1}{n-p-1}. \qquad (6)$$
In the above equation (6), $R^2$ stands for coefficient of determination, $p$ is number of descriptors and $n$ is the total
number of basins.
Due to large number of descriptors used in our analysis there is always a possibility of mutual correlation between
descriptors. To identify this mutual correlation between descriptor, VIF test is put to service. A cutoff value of 5 is
used below which a selected model is classified as "inutilizable" (Ganora et al., 2009; Gallice et al., 2015).
The selected models are further tested for average error generation ($\Delta$) in the overall workspace framed by the
descriptors constituting the models. The error test is carried out by assuming one station at a time as an ungauged
and removing its descriptor and hydrological data from the database. Afterwards models are recalibrated to estimate
the unknown flow regimes by using k-nearest neighbors ($KNN$) algorithm which relies on the selection of optimum



numbers of $NNs$ of $u_g$. The selection of appropriate number of unique $NNs$ is an important step in the procedure,
because too small number of neighbors can result in over simplification of results; while too many neighbors may
cause error in the final results. Following the procedure proposed by Samaniego et al. 2010, we opted for 5 $NNs$
after thoroughly scrutinizing from 1 to 9 (for details please refer to Samaniego et al., 2010). The unique $NNs$ in the
distance-based workspace are defined as the ones having distinct descriptive values. With workspace formulated by
multiple descriptors, the duplication in any of the descriptor values especially for the basins positioned near $u_g$, will
result in adding extraneous (or junk) variable to the predicting model resulting in inflated standard errors. The
singularity in descriptor values ensures that the dissimilarity between the basins is evenly shared by the descriptors
developing the predictive model. Furthermore, many basins having same descriptor values make it difficult to
nominate predefined number of $NNs$ of $u_g$.
The obtained results are compared with the original flow regimes to acquire the value of total dissimilarity
magnitude ($D_T$). The test, in totality, requires extraordinary computation power owing to the involvement of a
number of statistical operations. To minimize the computational burden, only a limited number of regression models
having, comparatively, good $R^2_{adj}$ values, are used to execute the regional regimes. The overall error ($\Delta$) for each
model (classified as having a better $R^2_{adj}$ value) is deduced by the following equation (7);
$$\Delta = \frac{\sum_{k=1}^{n-u_g}|D_T = f(Q_{k,act}, Q_{k,sim})|}{(n-1)} \tag{7}$$
where $D_T$ defines the total dissimilarity between the actual ($Q_{act}$) and simulated ($Q_{sim}$) regimes and the index $k$
expresses the station number.
The application of equations (6) and (7) to execute $R^2_{adj}$ and $\Delta$ values, respectively, is trivial in the selection of $OM$.
The model with comparatively higher $R^2_{adj}$ and least $\Delta$ value is selected to make initial estimation. However, the
implementation of $OM$ to the entire study area is always argued as problematic owing to the dynamic hydrological
response of basins to the changing descriptors. Besides extensive research done in the field of predictive hydrology,
hydrological response of basins could never be precisely quantified against the basin characteristics. The primary
advantage of using distance-based model workspace is that it can suggest an alternative workspace to counter the
issue of generalization due to the extension of $OM$ to the overall study area thus suggesting an appropriate
workspace for the prediction of hydrological data even at the localized level (for individual basin). We intend to
improve the estimates of the $OM$ by swapping it with another model, called Swapped model ($SM$), under the
predefined criteria. The predefined criteria include examining $R^2_{adj}$ and $\Delta$ values of the $OM$ and $SM$ for close-
proximity. The term "close-proximity" (or "almost similarity") in global performance is defined by, not more than
10% variation in $R^2_{adj}$ and $\Delta$ values of $OM$ and $SM$ (Qamar et al., 2016). The criteria are not strict in intrinsic sense.
However, the higher variation allowance will increase the risk of increased localized error. Whereas, allowing lower
variation will further complicate the selection of $SM$.



**5 Model swapping: logic, assumption, and implementation**

The alternative space is selected under the hypothesis that the $u_g$ and its $NNs$ form a unique region of influence (ROI) (Korn and Muthukrishnan, 2000). Inside ROI, the orientation of $u_g$ among its $NNs$ and the average error ($\Delta_{NN}$) generated in the estimation of hydrological data of $NNs$ of $u_g$ can act as comparative performance indicators of the alternative model space against the originally selected model space.

The application of model swapping for the improvement of predicted hydrological regime at $u_g$ commences by splitting the workspace of $OM$ around $u_g$ into six equal sectors (see Figure 4). The number of sectors occupied by $NNs$ of $u_g$ are counted to define a so-called coverage factor ($C_f^{OM}$). Afterwards, the hydrological data of each $NN$ of $u_g$ is predicted to estimate average error ($\Delta_{NN}^{OM}$) as defined by equation (7), in the ROI of $u_g$. The factor $\Delta_{NN}$ is useful in the sense that it transpires the model performance in the localized area containing $u_g$. The same parameters ($C_f^{SM}$ and $\Delta_{NN}^{SM}$) are estimated for the workspace of $SM$. The statistical results of $SM$ are accepted, if and only if $C_f^{SM} > C_f^{OM}$ and $\Delta_{NN}^{SM} < \Delta_{NN}^{OM}$.

The hydrological data of $NNs$ of $u_g$ in descriptors space are averaged to acquire the flow regime. By definition, the executed mean for $u_g$ will always be located in the middle of its $NNs$. The transformation of descriptive data to hydrological data is more meaningful if the same location pattern is actually depicted by the descriptive values of $u_g$ and its $NNs$. Broadly speaking, the actual location of $u_g$ in descriptors space should, ideally, overlap or align closely to the center formed by the mean of descriptors values of its $NNs$ (see Figure 4).

**Figure 4**

For example, referring to the Figure (4), the mean of hydrological data of $NNs$ of $u_g$ in the workspace of the models $(D_a, D_b)$ and $(D_c, D_d)$ is always converged to the center ($H_C^1$ and $H_C^2$ respectively). Whereas, the actual position of $u_g$ in the workspace formed by $(D_c, D_d)$ is closer to the virtual center formed by the descriptive values of its $NNs$ as compared to that of $(D_a, D_b)$. Therefore, the workspace $(D_c, D_d)$, in comparative terms, better satisfies the condition of meaningful transformation. Whereas, $u_g$ is ideally located in $(D_e, D_f)$ owing to the overlapping of its hypothetical and actual positions in the given workspace. The selected workspace is further tested for the localized error generation ($\Delta_{NN}$) by estimating hydrological data of $NNs$ of $u_g$ and computing average error by utilizing equation (7) in ROI of $u_g$.

It should be noted that with almost similar error magnitude in the overall workspace ($\Delta$), the lower magnitude of $\Delta_{NN}$ ensures the better prediction ability (with lower error) of the $SM$ in the localized area containing $u_g$. Although the application of $KNN$ is straight forward but it has been severely criticized for not taking into the account, the descriptive dissimilarity (or distance) between the selected $NNs$ and $u_g$ by allocating equal weightage to the selected neighbors. To address the stated problem in $KNN$, Hechenbichler and Schliep (2004) proposed a weighted coefficient to increase the weightage of closer neighbor in the estimating hydrological data of $u_g$ basin. Since the



effect of descriptors on the river flows varies unpredictably over a shorter distance, no standard method exists in
literature for the quantification of error magnitude per unit increase in distance (or dissimilarity) between the basins,
therefore, the method is not applicable for the proposed methodology. However, the location of $u_g$ in the middle of
its $NNs$ ensures the equitable distance of each $NN$ from $u_g$ and hence legitimizing equal weightage for each $NN$.
The proposed methodology is carried out in the $R$ statistical environment. The technique is very useful because non-
monotonic functions like rainfall can be introduced with a scalar descriptor to define suitable workspace for the
selection of $NNs$.
**6 Results and Discussion**
Following the procedure outlined for the selection of most appropriate model, we enlist the models, in Table (2),
which fulfilled the set criteria. The model with lower $\Delta$ value and higher $R_{adj}^2$ value, nominated as an $OM$, is used
for the assessment of hydrological data in an $u_g$. Within the workspace of $OM$, the flow regimes of predefined
number of $NNs$ of $u_g$ are averaged to predict the hydrological regime of $u_g$.
**Table (2)**
The descriptive models in Table (2) are constituted by 2-descriptors. The previous research works have shown that
the increased number of descriptors in the predictive model will increase the efficiency of the model output
(Kjeldsen and Jones, 2009; Kjeldsen et al., 2014). However, due to computational limitations, we opted to execute
the results by using models with 2-descriptors.
Out of numerous diverse descriptors used in our work, the climatic and geomorphological descriptors constituted the
most suitable models for the prediction. More specifically, the model constituted by ($quota\_media$,
$fa70percento$) is used for the initial estimations about hydrological data at $u_g$. The defined model evaluation
parameters viz; $R_{adj}^2$ and $\Delta$ equaled 0.291 and 0.660, respectively. The formation of better predictive models by
climatic and geomorphological descriptors is in line with the typology of the study area containing the selected
basins. For example, the descriptor ($fa70percento$) which is one of the constituent descriptor in the selected
models is relevant because of its strong influence on the basin response in the mountainous study area. Whereas, the
dominating geographical descriptor ($quota\_media$) maintains its significance by providing a synthetic explanation
of flow pattern. The methodology, thus, not only gives us luxury of simulating complicated flow regimes while
maintaining significance of peak discharge with fewer descriptors but also explains a logical connection between
flow magnitudes and selected descriptors.
The values of $\Delta_{NN}$ and $C_f$ for the selected $OM$ and $SM$ for 124 stations are summed up in Figure (5);
**Figure (5)**
The above figure suggests the response of 124 stations against the set criteria of model swapping. It is worth
mentioning that the essence of entire distance-based methodology is the quantification of dissimilarity between





269 basins in numeric terms. Occasionally, the descriptive values execute zero dissimilarity between the basins due to

270 absolute similarity, which results in the concentration of descriptors' values at a particular section of the workspace

271 thus creating a hardship in nominating the unique $NNs$ of $u_g$. Therefore, the selected models (both $OM$ and $SM$) are

272 further tested to check degree of scatterness of their values. The descriptive values arranged in ascending order are

273 plotted (against the station number) to check the uniqueness by observing the entire plot for the horizontal

274 section(s), which represent similarity in the descriptors' values. The test will ensure that the frequency ($\mathcal{F}$) of each

275 descriptor value ($d_i$) is equal to one ($i.e., \mathcal{F}d_i = 1$) resulting in the uniform distribution of $d_i$ over the model

276 workspace. The plots generated for each dominating descriptor to check the degree of scatterness are sketched in

277 Figure (6);

278          **Figure (6)**

279 The above figure clearly states that apart from descriptors ($clc_3$ and $delta\_mese$), the desired degree of scatterness

280 is obtained for the remaining descriptors. Therefore, the enlisted models containing one of ($clc_3$ and $delta\_mese$)

281 are sieved out due to difficulty in nominating a unique $NN$ of $u_g$.

282 Eventually, after satisfying all the formalities, the selected $SM$ are ultimately exercised for the statistical

283 improvement of the prediction. The results for 45 stations are compared in Table (3) by using performance indexes

284 such as Root Mean Square Error ($R$), Nash-Sutcliffe Efficiency ($N$), and Mean Absolute Error ($M$). On average $SM$

285 produced lesser error than the $OM$.

286          **Table (3)**

287 The results in Table (3) are the best examples to interpret the effectiveness of underlying assumptions of statistical

288 improvement of hydrological data by creating better spatial coverage and reducing the neighboring error around $u_g$.

289 For example, the output of stations 90 and 95 are significantly improved after swapping the $OM$ with the $SM$ due to

290 the comprehensive fulfillment of the set criteria for model swapping. Whereas, for stations 9 and 15 the results are

291 marginally elevated due to border line contentment of the swapping criteria. It can further be noted that the present

292 methodology provides comparatively better results when served with model based on climatic-geomorphologic

293 descriptors while the land use descriptors execute the least accurate results. The reason lies in the fact that the flow

294 magnitudes are directly dependent on the climatic-geomorphologic descriptors, while land use descriptors have

295 comparatively lesser effect on the magnitude of flow and occurrence of peak flows in the study area (Confortola et

296 al. 2013).

297 During the dissimilarity measurement between the flow regime, the peak flow position and magnitude are given

298 specific importance by introducing $L_{sp}$ and $V_{sp}$. Therefore, the prediction abilities are further explored to measure

299 the efficiency of the peak flow position w.r.t time and are elaborated in Table (4);

300          **Table (4)**



The monthly difference of "zero" represents the exact temporal estimation of the peak flow. Whereas, the values
greater than "zero" indicates the monthly temporal difference between the predicted and actual peak. For example,
the monthly difference of 2 indicates that the peak flow is estimated two months prior or post the occurrence of peak
flow in actual regime. It can clearly be noted that $SM$ better predicts the peak flow w.r.t time as compared to $OM$,
which misses it more frequently.
It should be noted that the proposed methodology only provides a comparative performance signature for the
prediction of flow regimes at $u_g$. The procedure comprehensively defines the comparative performance of 2-models
($OM$ and $SM$) beforehand by thoroughly investigating $C_f$ and $\Delta_{NN}$. It should also be borne in mind that the
procedure does not give any numeric value about the model performance indices ($R, N$ and $M$) in advance, however
it definitely identifies the better predictive model, statistically. This unique ability makes it an ideal tool for the use
in hydrological data prediction.
Although the output of prediction is more efficient using newly developed technique, however the result obtained
for station (82), are comparatively weaker than the $OM$ besides the fulfillment of swapping criteria for $SM3$. The
obvious reason, of deviation from the expected output, seems to be the simplified approach which is followed to
execute the error magnitude in the overall workspace and cluster (constituted by $u_g, u_g^{NN}$, and $NN$s of $u_g^{NN}$).
However, the issue can be effectively addressed by studying the change in error magnitude per unit change in
distance between the stations, which is ignored in our work. Moreover, it can be argued that the criteria defined for
model swapping is tough owing to which only 36% of the total basins could satisfy it. Nevertheless, with increasing
availability of meaningful descriptors around the globe, the proposed technique will become more effective. The
methodology holds a wide application spectrum in the fields of water management, flow trend analysis,
reconstitution of hydrological regimes, and temporal-and-magnitudinal prediction of peak discharge.
**7 Conclusion**
In this study, the distance matrices of descriptors and hydrological data are estimated and linked through regression
modelling to identify the most effective descriptive models. Afterwards, based on the values of $R_{adj}^2$ and $\Delta$,
statistically most feasible model is selected. The dissimilarity based-regionalization model is then coupled with
$KNN$ method to constitute the model space for initial predictions of flow regimes. The predicted results are then
improved by swapping it with another model having similar global performance.
The aims of changing the workspace of $u_g$ are; to have the better orientation of $u_g$ among its $NNs$ to increase the
coverage factor, and to reduce $\Delta_{NN}$ in the cluster formed by $u_g, u_g^{NN}$ and the $NN$s of $u_g^{NN}$. Once the defined criteria
are fulfilled, $SM$ is used to produce the flow regimes. The statistical performance parameters in terms of $R, N$ and $M$
evaluated for $SM$ are better than the $OM$. It is, however, not easy to fulfill the set requirements of model swapping
due to difficulty in orientating $u_g$ in the middle of it $NNs$ while ensuring lower $\Delta_{NN}^{SM}$ than $\Delta_{NN}^{OM}$. Nevertheless, with
extensive research on the field of hydrology coupled with the identification, execution and availability of more





meaningful catchment descriptors, the application of the proposed methodology is expected to become straight
forward.
The approach followed an unorthodox signature rule that gives an option to identify the basin-specific best
predictive model instead of having a generalized predictive model for the whole study area. Alongside that, it also
gives provision for the temporal estimation of the peak discharge magnitude. These properties make it an ideal tool
to be used in field of predictive hydrology and climatology.
**Acknowledgments:**
This Publication was funded by the German Research Foundation (DFG) and the University of Wuerzburg through
the Open-Access Publishing Programme. The complex simulations were performed in the Computation Lab of
Department of Energy System Engineering, University of Agriculture, Faisalabad, by the assistance of Dr. Waseem
Amjad.

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



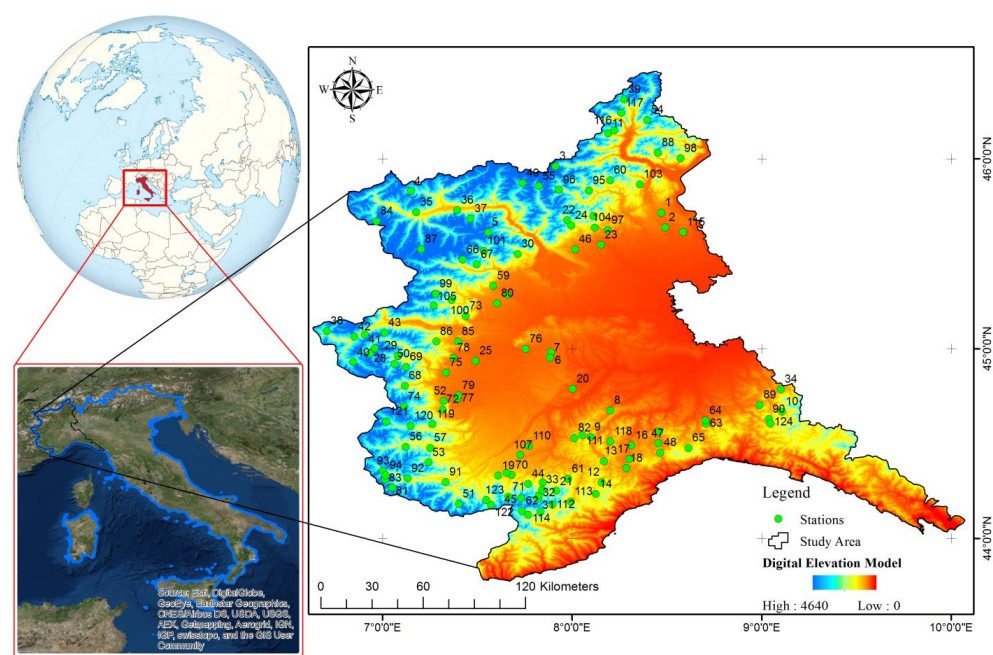

**Figure 1: Location of gauging stations used in the analysis (Source: Qamar et al., 2018).**





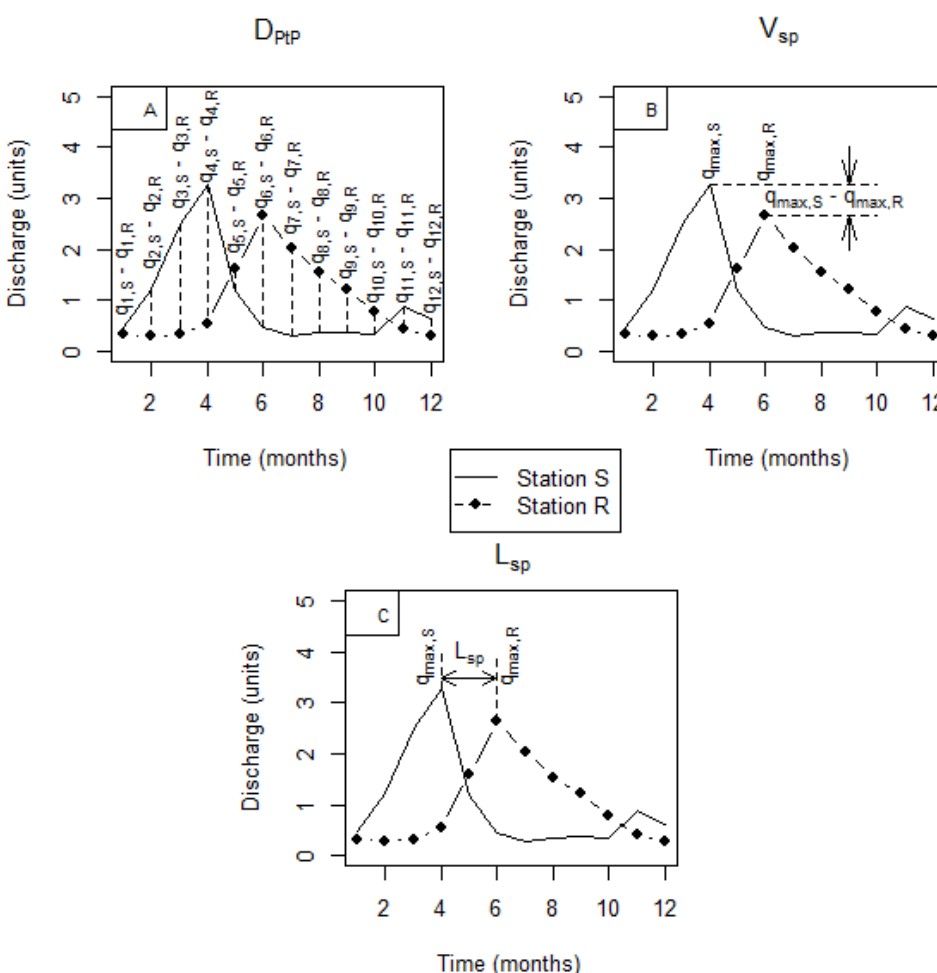

**Figure 2: Diagrammatic representation of types of dissimilarities used in our work.**



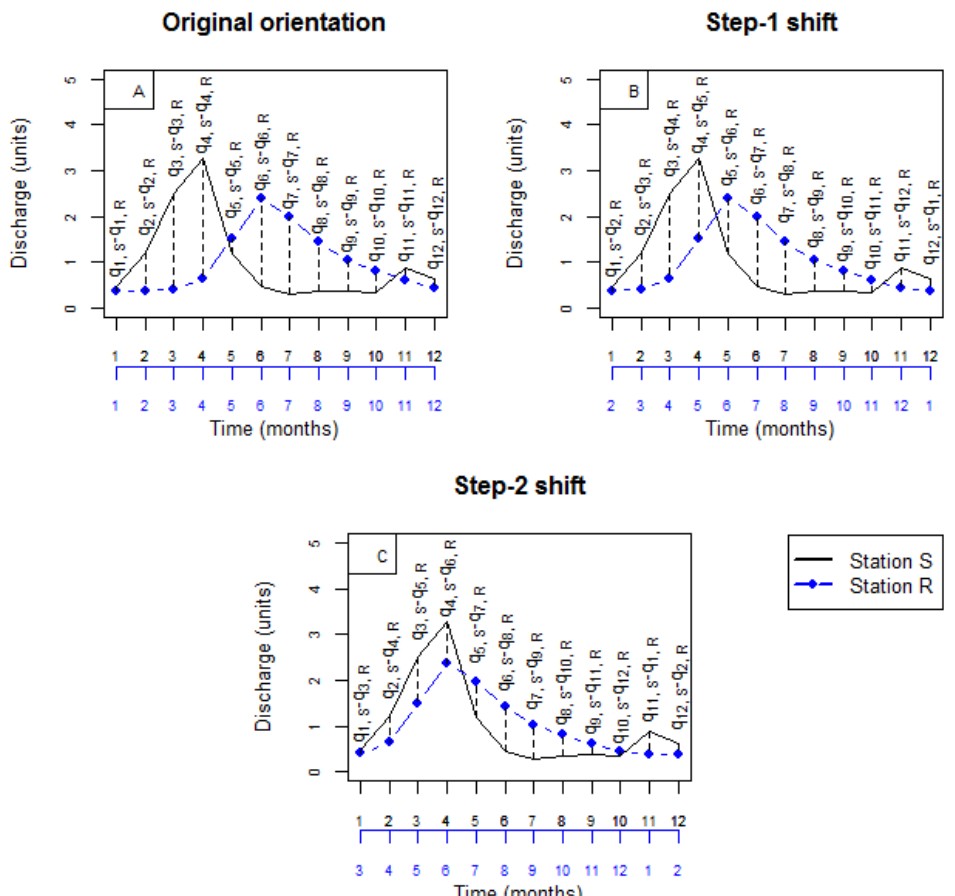

**Figure 3: Step wise shifting of peak R towards S.**





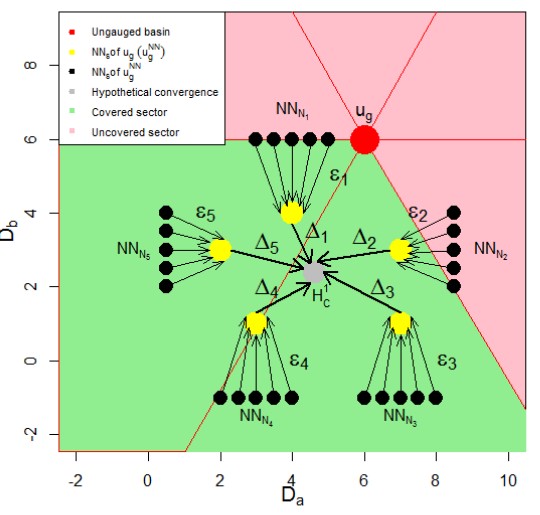

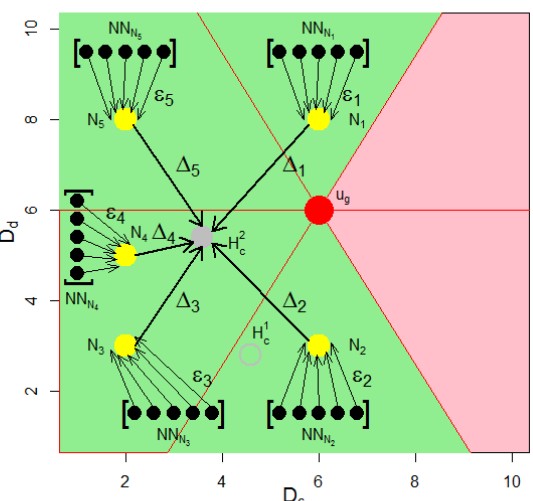

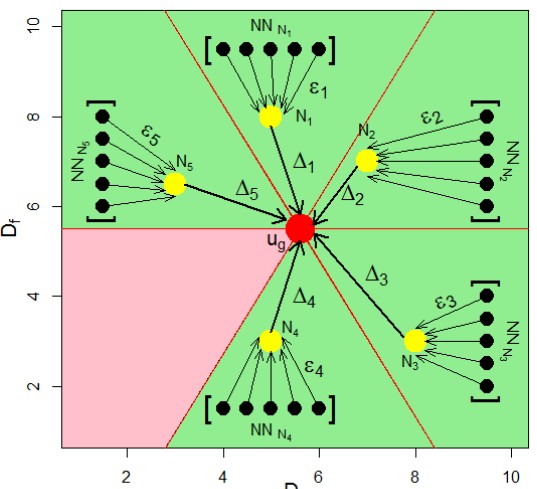

Figure 4: Arrangement of $u_g$ among its NNs in the workspace constituted by $(D_a, D_b)$, $(D_c, D_d)$ and $(D_e, D_f)$.
The preference order from highest to lowest is $(D_e, D_f)$, $(D_c, D_d)$ and $(D_a, D_b)$.





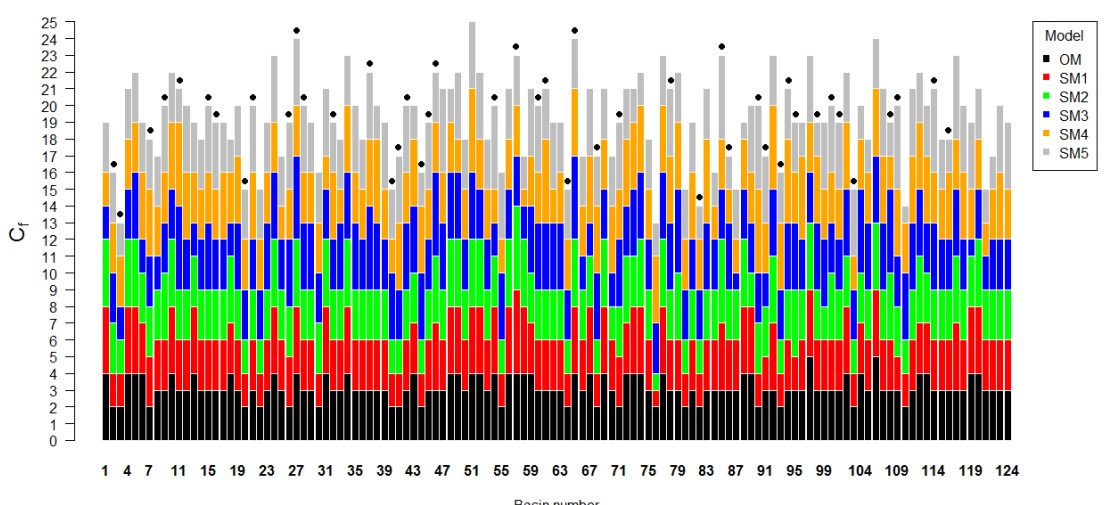

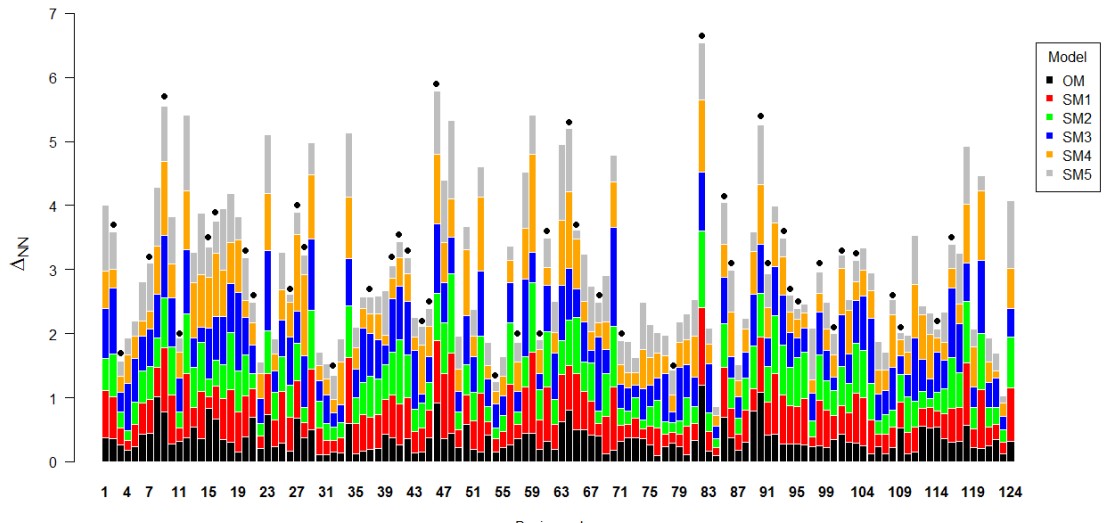

**Figure 5: Analyzing $C_f$ and $\Delta_{NN}$ values against the set criteria of model swapping. The black dots above the bars**
**plots represent the stations where the set criteria of swapping are fulfilled.**





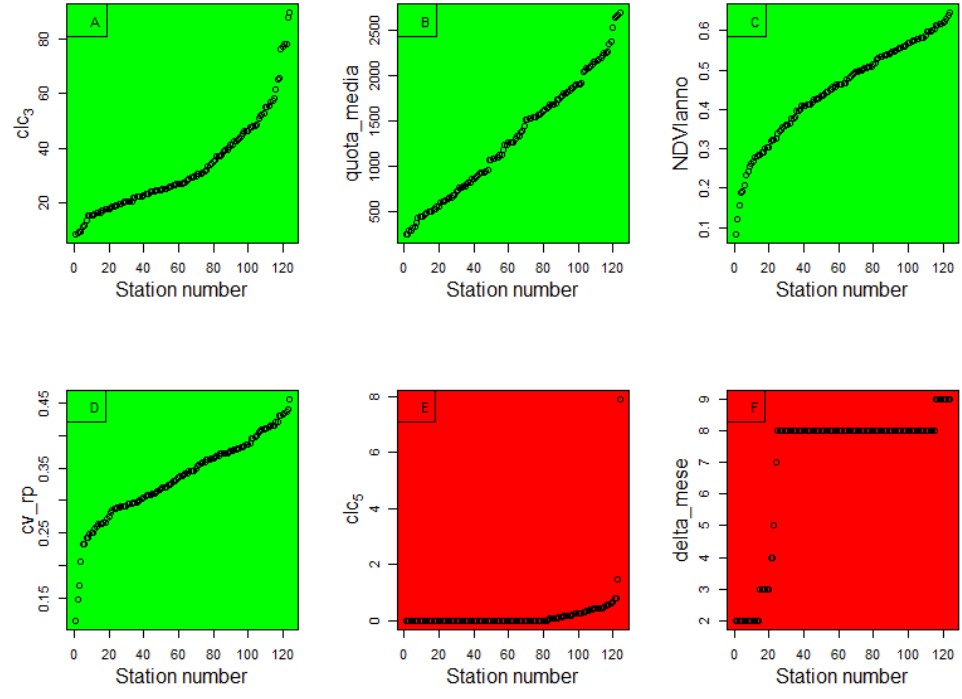

**Figure 6: Analyzing frequency of occurrence of descriptor values. The plots with green background represent**
**the descriptors having better degree of scatterness while the ones with red background could not show**
**uniqueness in descriptor values.**



Table 1: Maximum, mean and minimum values of the selected descriptors.

| Descriptor symbol | Descriptor definition | Descriptor values | | |
|---|---|---|---|---|
| | | Maximum | Mean | Minimum |
| $clc_3$ | Percentage area of the basin containing herbaceous vegetation, grass-grazing, special crops, olive groves, vineyards, crops | 89.32 | 33.16 | 8.57 |
| $quota\_media$ | Average Basin Elevation (m) | 2682 | 1306.52 | 244 |
| $a25percento$ | 25th percentile of the hypsographic curve | 3091 | 1637.10 | 274 |
| $fa70percento$ | 70th percentile of the width function | 208645 | 41397.39 | 5721 |
| $fa85percento$ | 85th percentile of the width function | 241407 | 47618.9 | 7325 |
| $fa90percento$ | 90th percentile of the width function | 264278 | 53484.19 | 8208 |
| $cn_3\_std$ | Standard deviation of Curve Number related to the moist soil | 32.34 | 9.87 | 2.24 |
| $sd\_rp$ | Standard deviation of the rainfall regime (mm) | 89.17 | 34.43 | 8.84 |
| $area\_bacino$ | Basin area (m$^2$) | 25640 | 1276.331 | 22 |
| $x\_baricentro$ | X-coordinate of the basin | 508450 | 401454.8 | 319450 |
| $y\_baricentro$ | Y-coordinate of the basin | 5129050 | 4977667 | 4886350 |
| $delta\_mese$ | Time interval between maximum and minimum monthly average of rainfall (months) | 9 | 7.056 | 2 |

Table 2: List of selected models with $R^2_{adj}$ and Δ values.

| Models | Symbolic representation | $R^2_{adj}$ | Δ | Percentage change in value w.r.t OM | |
|---|---|---|---|---|---|
| | | | | $R^2_{adj}$ | Δ |
| $quota\_media$, $fa70percento$ | OM | 0.2916 | 0.6602 | 0 | 0 |
| $quota\_media$, $fa85percento$ | SM1 | 0.2914 | 0.6848 | 0.069 | 3.726 |
| $quota\_media$, $fa90percento$ | SM2 | 0.2910 | 0.7020 | 0.206 | 6.331 |
| $a25percento$, $cn_3\_std$ | SM3 | 0.3198 | 0.7113 | 9.671 | 7.740 |
| $a25percento$, $sd\_rp$ | SM4 | 0.3070 | 0.7083 | 5.281 | 7.286 |
| $clc_3$, $quota\_media$ | SM5 | 0.2991 | 0.7214 | 2.572 | 9.270 |



**Table 3: Results executed by original and swapped models in terms of $R$, $N$ and $M$ along with $C_f$ and $\Delta_{NN}$ values. The bold numbers represent the models where swapping criteria are fulfilled.**

| St. No. | OM R | OM N | OM M | $\Delta^{OM}_{NN}$ | $C_f$ | SM1 R | SM1 N | SM1 M | $C_f$ | $\Delta^{SM}_{NN}$ | SM2 R | SM2 N | SM2 M | $C_f$ | $\Delta^{SM}_{NN}$ | SM3 R | SM3 N | SM3 M | $C_f$ | $\Delta^{SM}_{NN}$ | SM4 R | SM4 N | SM4 M | $C_f$ | $\Delta^{SM}_{NN}$ | SM5 R | SM5 N | SM5 M | $C_f$ | $\Delta^{SM}_{NN}$ |
|---|---|---|---|---|---|---|---|---|---|---|---|---|---|---|---|---|---|---|---|---|---|---|---|---|---|---|---|---|---|---|
| 2 | 0.326 | 0.326 | 3.516 | 0.359 | 2 | 0.326 | 0.326 | 3.516 | 2 | 0.657 | 0.326 | 0.326 | 3.516 | 3 | 0.663 | 0.281 | 0.499 | 2.667 | 3 | 1.040 | 0.336 | 0.287 | 2.608 | 3 | 0.288 | 0.314 | 0.374 | 3.049 | 3 | 0.588 |
| 3 | 0.199 | 0.954 | 1.862 | 0.270 | 2 | 0.199 | 0.954 | 1.862 | 2 | 0.257 | 0.199 | 0.954 | 1.862 | 2 | 0.374 | 0.199 | 0.954 | 1.862 | 2 | 0.305 | 0.128 | 0.981 | 1.248 | 3 | **0.246** | 0.499 | 0.710 | 4.686 | 2 | 0.238 |
| 7 | 0.263 | 0.821 | 2.207 | 0.442 | 3 | 0.233 | 0.859 | 1.875 | 3 | 0.526 | 0.233 | 0.859 | 1.875 | 3 | 0.525 | 0.271 | 0.810 | 2.329 | 5 | 0.584 | 0.260 | 0.844 | 2.120 | 4 | **0.269** | 0.276 | 0.802 | 2.040 | 3 | 0.761 |
| 9 | 0.712 | 0.390 | 6.188 | 0.772 | 3 | 0.649 | 0.492 | 5.687 | 3 | 1.008 | 0.649 | 0.492 | 5.687 | 4 | **0.770** | 0.883 | 0.062 | 8.139 | 3 | 0.976 | 0.198 | -0.04 | 7.583 | 3 | 1.152 | 0.641 | 0.505 | 5.080 | 4 | 0.866 |
| 11 | 0.317 | 0.774 | 2.976 | 0.313 | 3 | 0.317 | 0.774 | 2.976 | 3 | 0.212 | 0.320 | 0.769 | 2.999 | 3 | 0.254 | 0.197 | 0.912 | 1.772 | 5 | 0.525 | 0.406 | 0.588 | 3.561 | 5 | **0.289** | 0.260 | 0.848 | 2.285 | 2 | 0.213 |
| 15 | 0.360 | 0.675 | 2.907 | 0.840 | 3 | 0.517 | 0.332 | 4.688 | 3 | 0.181 | 0.517 | 0.332 | 4.688 | 3 | 0.266 | 0.315 | 0.752 | 2.384 | 4 | **0.804** | 0.296 | 0.781 | 1.752 | 3 | 0.788 | 0.296 | 0.781 | 2.525 | 4 | **0.474** |
| 16 | 0.233 | 0.847 | 2.301 | 0.672 | 3 | 0.233 | 0.847 | 2.301 | 3 | 0.512 | 0.433 | 0.473 | 4.555 | 3 | 0.398 | 0.228 | 0.853 | 1.689 | 3 | 0.684 | 0.167 | 0.922 | 1.504 | 3 | **0.360** | 0.169 | 0.920 | 1.500 | 4 | **0.506** |
| 20 | 0.299 | 0.423 | 3.143 | 0.387 | 2 | 0.299 | 0.423 | 3.143 | 2 | 0.642 | 0.299 | 0.423 | 3.143 | 3 | 0.643 | 0.277 | 0.505 | 2.573 | 3 | 0.578 | 0.269 | 0.534 | 2.505 | 3 | **0.264** | 0.331 | 0.295 | 3.346 | 3 | 0.678 |
| 21 | 0.509 | 0.423 | 5.705 | 0.700 | 3 | 0.441 | 0.566 | 4.965 | 3 | 0.435 | 0.509 | 0.423 | 5.705 | 2 | 0.243 | 0.212 | 0.900 | 2.154 | 3 | 0.439 | 0.332 | 0.754 | 3.484 | 4 | **0.360** | 0.268 | 0.840 | 2.316 | 4 | **0.317** |
| 26 | 0.742 | 0.645 | 6.830 | 0.171 | 2 | 0.707 | 0.678 | 6.376 | 3 | 0.530 | 0.715 | 0.671 | 6.296 | 3 | 0.499 | 0.735 | 0.652 | 6.503 | 4 | 0.754 | 0.655 | 0.724 | 5.358 | 4 | 0.535 | 0.696 | 0.688 | 5.482 | 4 | **0.120** |
| 27 | 0.135 | 0.968 | 1.274 | 0.684 | 4 | 0.135 | 0.968 | 1.274 | 4 | 0.579 | 0.135 | 0.968 | 1.274 | 3 | 0.586 | 0.114 | 0.977 | 1.057 | 5 | **0.500** | 0.309 | 0.834 | 2.663 | 3 | 1.202 | 0.180 | 0.944 | 2.016 | 4 | 0.343 |
| 28 | 0.292 | 0.895 | 2.609 | 0.368 | 3 | 0.292 | 0.895 | 2.609 | 3 | 0.237 | 0.292 | 0.895 | 2.609 | 3 | 0.241 | 0.254 | 0.921 | 2.342 | 4 | 0.813 | 0.231 | 0.934 | 2.000 | 3 | 0.265 | 0.265 | 0.913 | 2.621 | 4 | **0.308** |
| 32 | 0.234 | 0.913 | 2.404 | 0.152 | 3 | 0.234 | 0.913 | 2.404 | 3 | 0.181 | 0.234 | 0.913 | 2.404 | 3 | 0.205 | 0.222 | 0.922 | 2.078 | 3 | 0.224 | 0.237 | 0.911 | 2.384 | 4 | **0.142** | 0.342 | 0.815 | 3.461 | 3 | 0.377 |
| 37 | 0.457 | 0.508 | 3.880 | 0.191 | 3 | 0.405 | 0.613 | 3.456 | 3 | 0.498 | 0.290 | 0.803 | 2.293 | 3 | 0.652 | 0.298 | 0.791 | 2.607 | 5 | 0.666 | 0.323 | 0.754 | 3.125 | 4 | 0.300 | 0.284 | 0.810 | 2.816 | 4 | **0.172** |
| 40 | 0.147 | 0.965 | 1.526 | 0.375 | 2 | 0.147 | 0.965 | 1.526 | 2 | 0.661 | 0.147 | 0.965 | 1.526 | 2 | 0.667 | 0.159 | 0.959 | 1.631 | 4 | 0.839 | 0.200 | 0.936 | 1.911 | 3 | 0.322 | 0.142 | 0.973 | 1.460 | 3 | **0.199** |
| 41 | 0.318 | 0.521 | 3.189 | 0.263 | 3 | 0.282 | 0.623 | 2.785 | 2 | 0.641 | 0.343 | 0.441 | 3.223 | 3 | 0.999 | 0.238 | 0.731 | 2.188 | 3 | 0.841 | 0.238 | 0.731 | 2.335 | 4 | 0.442 | 0.318 | 0.519 | 3.117 | 4 | **0.248** |
| 42 | 0.356 | 0.825 | 3.259 | 0.365 | 3 | 0.356 | 0.825 | 3.259 | 3 | 0.638 | 0.356 | 0.825 | 3.259 | 3 | 0.665 | 0.232 | 0.926 | 2.253 | 4 | 0.839 | 0.298 | 0.877 | 2.796 | 3 | 0.428 | 0.248 | 0.915 | 2.471 | 3 | **0.251** |
| 44 | 0.263 | 0.836 | 3.084 | 0.155 | 2 | 0.263 | 0.836 | 3.084 | 2 | 0.378 | 0.263 | 0.836 | 3.084 | 2 | 0.379 | 0.256 | 0.845 | 2.544 | 4 | **0.148** | 0.220 | 0.885 | 1.896 | 3 | 0.765 | 0.219 | 0.887 | 2.117 | 2 | 0.286 |
| 45 | 0.233 | 0.929 | 2.309 | 0.376 | 3 | 0.233 | 0.929 | 2.309 | 3 | 0.431 | 0.233 | 0.929 | 2.309 | 3 | 0.430 | 0.333 | 0.855 | 3.240 | 3 | 0.410 | 0.223 | 0.935 | 2.166 | 3 | 0.467 | 0.185 | 0.955 | 1.633 | 4 | **0.275** |
| 46 | 0.897 | -1.28 | 9.007 | 0.917 | 3 | 0.897 | -1.28 | 9.007 | 3 | 0.969 | 0.897 | -1.28 | 9.007 | 4 | 0.984 | 0.583 | 0.039 | 5.638 | 5 | **0.88** | 0.754 | -0.61 | 7.605 | 3 | 1.076 | 0.716 | -0.45 | 7.284 | 3 | 0.988 |
| 54 | 0.432 | 0.407 | 3.444 | 0.148 | 4 | 0.432 | 0.407 | 3.444 | 4 | 0.213 | 0.515 | 0.157 | 4.435 | 3 | 0.166 | 0.384 | 0.533 | 3.724 | 2 | 0.375 | 0.543 | 0.066 | 5.214 | 2 | 0.200 | 0.390 | 0.516 | 3.207 | 5 | **0.147** |
| 57 | 0.334 | 0.607 | 3.355 | 0.360 | 3 | 0.336 | 0.602 | 3.302 | 5 | **0.220** | 0.331 | 0.614 | 3.310 | 4 | **0.204** | 0.337 | 0.600 | 3.288 | 3 | 0.311 | 0.431 | 0.345 | 4.381 | 4 | 0.462 | 0.382 | 0.486 | 3.941 | 3 | 0.301 |
| 60 | 0.361 | 0.508 | 4.042 | 0.196 | 3 | 0.361 | 0.508 | 4.042 | 3 | 0.461 | 0.361 | 0.508 | 4.042 | 3 | 0.465 | 0.367 | 0.492 | 3.371 | 4 | 0.247 | 0.267 | 0.731 | 2.653 | 4 | 0.376 | 0.297 | 0.667 | 2.861 | 3 | **0.161** |
| 61 | 0.551 | 0.555 | 5.102 | 0.316 | 3 | 0.551 | 0.555 | 5.102 | 3 | 0.854 | 0.551 | 0.555 | 5.102 | 3 | 0.857 | 0.534 | 0.583 | 5.035 | 3 | 0.730 | 0.427 | 0.733 | 4.072 | 5 | **0.282** | 0.355 | 0.816 | 3.293 | 3 | 0.451 |
| 64 | 0.391 | 0.694 | 3.725 | 0.809 | 2 | 0.391 | 0.694 | 3.725 | 2 | 0.696 | 0.391 | 0.694 | 3.725 | 2 | 0.703 | 0.313 | 0.803 | 2.912 | 3 | **0.789** | 0.488 | 0.522 | 4.941 | 3 | 1.194 | 0.472 | 0.552 | 4.900 | 3 | 0.990 |
| 65 | 0.324 | 0.653 | 3.226 | 0.496 | 4 | 0.324 | 0.653 | 3.226 | 4 | 0.875 | 0.324 | 0.653 | 3.226 | 4 | 0.879 | 0.276 | 0.747 | 2.668 | 5 | **0.445** | 0.376 | 0.531 | 3.695 | 5 | 0.782 | 0.645 | -0.38 | 6.576 | 3 | 0.138 |
| 68 | 0.438 | -1.16 | 4.124 | 0.407 | 2 | 0.458 | -1.35 | 4.351 | 2 | 0.191 | 0.485 | -1.64 | 4.675 | 3 | 0.201 | 0.530 | -2.16 | 4.919 | 2 | 1.153 | 0.479 | -1.57 | 4.536 | 4 | 0.519 | 0.407 | -.86 | 4.051 | 3 | **0.316** |
| 71 | 0.222 | 0.920 | 2.290 | 0.315 | 2 | 0.273 | 0.880 | 2.532 | 2 | 0.249 | 0.273 | 0.880 | 2.532 | 3 | 0.374 | 0.448 | 0.676 | 3.980 | 4 | 0.368 | 0.161 | 0.958 | 1.435 | 3 | **0.307** | 0.334 | 0.873 | 2.714 | 4 | 0.384 |
| 78 | 0.306 | 0.801 | 2.413 | 0.296 | 3 | 0.306 | 0.801 | 2.413 | 3 | 0.164 | 0.306 | 0.801 | 2.413 | 3 | 0.156 | 0.279 | 0.835 | 2.344 | 4 | **0.155** | 0.297 | 0.813 | 2.359 | 4 | **0.275** | 0.321 | 0.781 | 2.558 | 4 | 0.382 |
| 82 | 0.612 | 0.604 | 5.479 | 1.193 | 3 | 0.612 | 0.604 | 5.479 | 4 | 1.216 | 0.612 | 0.604 | 5.479 | 5 | 1.189 | 0.891 | 0.161 | 8.537 | 3 | **0.926** | 0.596 | 0.624 | 5.421 | 3 | 1.125 | 0.640 | 0.567 | 5.478 | 5 | 0.887 |
| 85 | 0.647 | 0.334 | 6.524 | 0.693 | 4 | 0.637 | 0.355 | 6.453 | 4 | 0.783 | 0.600 | 0.428 | 5.916 | 5 | **0.681** | 0.730 | 0.153 | 7.336 | 3 | 0.729 | 0.509 | 0.589 | 4.861 | 3 | 0.513 | 0.746 | 0.117 | 7.538 | 5 | 0.745 |
| 86 | 0.292 | 0.799 | 2.817 | 0.379 | 3 | 0.261 | 0.839 | 2.365 | 3 | 0.462 | 0.261 | 0.839 | 2.365 | 3 | 0.463 | 0.217 | 0.889 | 2.131 | 4 | **0.333** | 0.405 | 0.613 | 4.129 | 2 | 0.697 | 0.232 | 0.873 | 2.093 | 2 | 0.661 |
| 90 | 0.493 | 0.276 | 5.236 | 1.088 | 2 | 0.493 | 0.276 | 5.236 | 2 | 0.854 | 0.480 | 0.312 | 5.158 | 3 | **0.691** | 0.469 | 0.343 | 4.791 | 3 | **0.769** | 0.287 | 0.755 | 2.992 | 5 | **0.922** | 0.389 | 0.549 | 3.633 | 4 | **0.935** |
| 91 | 0.201 | 0.924 | 2.123 | 0.421 | 3 | 0.232 | 0.898 | 2.382 | 3 | 0.505 | 0.208 | 0.918 | 2.248 | 3 | 0.568 | 0.164 | 0.949 | 1.732 | 2 | 0.436 | 0.225 | 0.905 | 2.220 | 2 | **0.407** | 0.192 | 0.940 | 1.944 | 4 | **0.268** |
| 93 | 0.250 | 0.817 | 2.541 | 0.275 | 2 | 0.250 | 0.817 | 2.541 | 2 | 0.767 | 0.250 | 0.817 | 2.541 | 2 | 0.774 | 0.291 | 0.752 | 2.570 | 2 | 0.791 | 0.209 | 0.873 | 1.744 | 4 | 0.598 | 0.236 | 0.836 | 2.027 | 3 | **0.207** |
| 94 | 0.398 | 0.813 | 3.330 | 0.276 | 3 | 0.398 | 0.813 | 3.330 | 3 | 0.600 | 0.398 | 0.813 | 3.330 | 3 | 0.605 | 0.362 | 0.846 | 3.012 | 4 | 0.537 | 0.349 | 0.856 | 2.737 | 5 | 0.323 | 0.334 | 0.869 | 2.907 | 5 | **0.258** |
| 95 | 0.609 | 0.044 | 5.168 | 0.282 | 4 | 0.444 | 0.492 | 3.977 | 3 | 0.575 | 0.609 | 0.044 | 5.168 | 4 | 0.770 | 0.393 | 0.602 | 4.197 | 4 | **0.279** | 0.362 | 0.663 | 3.500 | 3 | 0.297 | 0.310 | 0.753 | 3.108 | 3 | **0.173** |
| 98 | 0.515 | -0.54 | 5.644 | 0.244 | 3 | 0.515 | -0.54 | 5.644 | 3 | 0.709 | 0.515 | -0.54 | 5.644 | 3 | 0.711 | 0.295 | 0.495 | 3.216 | 3 | 0.676 | 0.355 | 0.271 | 3.958 | 4 | **0.207** | 0.177 | 0.818 | 1.840 | 4 | 0.342 |
| 100 | 0.242 | 0.900 | 2.032 | 0.342 | 3 | 0.254 | 0.889 | 2.213 | 3 | 0.378 | 0.254 | 0.889 | 2.213 | 4 | 0.392 | 0.258 | 0.886 | 2.253 | 3 | 0.392 | 0.209 | 0.925 | 1.787 | 3 | 0.355 | 0.157 | 0.958 | 1.584 | 4 | **0.333** |
| 101 | 0.426 | 0.532 | 3.784 | 0.435 | 3 | 0.426 | 0.532 | 3.784 | 3 | 0.439 | 0.462 | 0.451 | 3.717 | 3 | 0.611 | 0.362 | 0.662 | 3.424 | 3 | 0.812 | 0.362 | 0.662 | 2.941 | 3 | 0.724 | 0.279 | 0.800 | 2.274 | 4 | **0.206** |
| 103 | 0.362 | 0.344 | 3.944 | 0.291 | 2 | 0.362 | 0.344 | 3.944 | 2 | 0.784 | 0.362 | 0.344 | 3.944 | 2 | 0.778 | 0.409 | 0.159 | 4.503 | 3 | 0.668 | 0.297 | 0.556 | 2.880 | 3 | 0.288 | 0.187 | 0.825 | 1.885 | 4 | **0.241** |
| 108 | 0.252 | 0.864 | 1.951 | 0.222 | 3 | 0.252 | 0.756 | 2.556 | 3 | 0.320 | 0.189 | 0.862 | 1.673 | 4 | **0.209** | 0.230 | 0.795 | 2.157 | 5 | **0.209** | 0.274 | 0.710 | 2.568 | 2 | 0.818 | 0.176 | 0.881 | 1.437 | 2 | 0.240 |
| 109 | 0.332 | 0.659 | 3.459 | 0.522 | 3 | 0.309 | 0.704 | 3.145 | 3 | 0.405 | 0.272 | 0.772 | 2.639 | 3 | 0.439 | 0.406 | 0.490 | 4.347 | 4 | 0.287 | 0.291 | 0.738 | 3.053 | 4 | **0.248** | 0.215 | 0.858 | 1.856 | 5 | **0.119** |
| 114 | 0.135 | 0.955 | 1.237 | 0.542 | 3 | 0.135 | 0.955 | 1.237 | 3 | 0.252 | 0.135 | 0.955 | 1.237 | 3 | 0.284 | 0.252 | 0.844 | 2.569 | 4 | 0.637 | 0.163 | 0.934 | 1.577 | 3 | 0.222 | 0.128 | 0.960 | 1.206 | 5 | **0.140** |
| 116 | 0.452 | -0.19 | 4.365 | 0.308 | 3 | 0.452 | -0.19 | 4.365 | 3 | 0.530 | 0.568 | -0.88 | 5.346 | 3 | 0.795 | 0.435 | -0.10 | 4.238 | 3 | 1.085 | 0.207 | 0.750 | 1.769 | 4 | **0.306** | 0.337 | 0.336 | 3.361 | 2 | 0.365 |



**Table 4: Peak flow prediction w.r.t. time by original and swapped models.**

| Station No. | Actual peak | OM | SM1 | SM2 | SM3 | SM4 | SM5 |
|---|---|---|---|---|---|---|---|
| 2 | 11 | 12 | - | - | - | 11 | - |
| 3 | 6 | 6 | - | - | - | 6 | - |
| 7 | 12 | 11 | - | - | - | 12 | - |
| 9 | 4 | 5 | - | 4 | - | - | - |
| 11 | 5 | 5 | - | - | - | 5 | - |
| 15 | 11 | 12 | - | - | 12 | - | 11 |
| 16 | 11 | - | - | - | - | - | 11 |
| 20 | 11 | 11 | - | - | - | 11 | - |
| 21 | 11 | 5 | - | - | - | 11 | 11 |
| 26 | 6 | 5 | - | - | - | - | 5 |
| 27 | 6 | 5 | - | - | 6 | - | - |
| 28 | 6 | 6 | - | - | - | - | 6 |
| 32 | 5 | 5 | - | - | - | 5 | - |
| 37 | 6 | 5 | - | - | - | - | 6 |
| 40 | 5 | 5 | - | - | - | - | 6 |
| 41 | 6 | 5 | - | - | - | - | 6 |
| 42 | 6 | 6 | - | - | - | - | 6 |
| 44 | 4 | 5 | - | - | 5 | - | - |
| 45 | 5 | 5 | - | - | - | - | 5 |
| 46 | 5 | 4 | - | - | 4 | - | - |
| 54 | 5 | 5 | - | - | - | - | 5 |
| 57 | 5 | 5 | 5 | 5 | - | - | - |
| 60 | 5 | 5 | - | - | - | - | 5 |
| 61 | 4 | 12 | - | - | - | - | 4 |
| 64 | 12 | 12 | - | - | 12 | - | - |
| 65 | 3 | 4 | - | - | 4 | - | - |
| 68 | 5 | 5 | - | - | - | - | 5 |
| 71 | 5 | 5 | - | - | - | 5 | - |
| 78 | 6 | 5 | - | - | - | 5 | - |
| 82 | 3 | 12 | - | - | 4 | 3 | - |
| 85 | 5 | 4 | - | 4 | - | - | - |
| 86 | 5 | 5 | - | - | 5 | - | - |
| 90 | 12 | 5 | - | 5 | 4 | 12 | 12 |
| 91 | 5 | 5 | - | - | - | - | 5 |
| 93 | 5 | 6 | - | - | - | - | 5 |
| 94 | 5 | 6 | - | - | - | - | 6 |
| 95 | 5 | 5 | - | - | 5 | - | 5 |
| 98 | 5 | 4 | - | - | - | 5 | - |
| 100 | 5 | 5 | - | - | - | - | 5 |
| 101 | 5 | 5 | - | - | - | - | 5 |
| 103 | 5 | 4 | - | - | - | - | 5 |
| 108 | 5 | 4 | - | 5 | - | - | - |
| 109 | 5 | 5 | - | - | - | 5 | 5 |
| 114 | 5 | 5 | - | - | - | - | 5 |
| 116 | 6 | 5 | - | - | - | 6 | - |