# Peer review of "Technical note: A novel technique to improve the hydrological estimates at ungauged basins by swapping workspaces"

_Hydrology and Earth System Sciences, 2018_

## Referee Comment (RC1) · Anonymous Referee #1 · 10 Oct 2018

Journal: Hydrology and Earth System Sciences (HESS)
Title: Technical note: A novel technique to improve the hydrological estimates at ungauged basins by swapping workspaces
MS No.: hess-2018-418
Article Iteration: First review

**Summary and Overall assessment**

The study aims to propose a new statistical technique to improve the predictive power of multiple linear regression models used for streamflow predictions in ungauged basins. The contribution of the technique, in essence, is based on defining new (dis)similarity indices—called point-to-point, lateral, and vertical distance measures—applied to monthly flow regimes. They have used monthly flow regimes, instead of using flow duration curves (FDCs), to compare the ungauged basin with its neighboring basins arguing that FDCs loss the time aspect of hydrologic data information content. Further, they seek to improve the flow estimation in the ungauged by "swapping" the original model (used for flow estimation) by other models *if* certain conditions are met; namely (1) increasing the covered space around the ungauged basin, and (2) reducing the overall model error. Authors developed their regression models based on climatic and geomorphologic descriptors of neighboring basins, and applied their framework across 124 basins in northwestern Italy.

In my evaluation the manuscript is **rejected**, with the kind encouragement of submitting a substantially revised version. The fundamental requirement of reporting a scientific research is not met, i.e. its clarity and testability/reproducibility. The wording and phrasing on many instances within the manuscript are short of a scientific text; needless to mention that the manuscript is not even proof-read. The manuscript is filled with unsubstantiated, poorly defined, and vague statements—instances are discussed within the section below. Parts of the discussions lack logical coherence and internal consistency (comments 2.f and 2.g). The work is not properly contextualized within the extant literature. The descriptors used in developing original models are not explained properly, and the results discussion lack a discussion on the uncertainties associated with the data used, developed method, as well as the discussion of results.

I encourage the authors to ask one or few experts in regionalization as well as a native English speaker to review their manuscript thoroughly before the re-submission.

**Major comments**

1. The manuscript is poorly structured.
   a. Parts of the Introduction (lines 64-82) are in fact methodology. Within the Introduction is important to discuss "why" your work matters and "why" the method is a contribution. That said, "how" the method works, should be discussed in the Methods section.
   b. The objective of the work is not clearly stated in the Introduction, instead mentioned at the end of the Study Area (lines 103-105). Further, you mentioned your primary focus is "accurate prediction". You need to exactly define what you mean by accuracy. If by accuracy you mean reducing bias, then you need to present the values of the models' biases to demonstrate the improvement (e.g. on table 3).

c. The manuscript does not have a Methods section. To better elaborate and emphasize your proposed method, I think it is essential to put sections 3-5 under an overarching section of Methods, consolidate them (some parts are hard to follow) and remove repetitions, and perhaps before section 2 Study Area. Further, the procedure explained within lines 21-221 (including the figure) is hard to follow. Please rephrase and explain more clearly.

d. Lines 93-95: you listed "some of the descriptors (out of 74 descriptors)" on table 1. Why did you choose these descriptors? Are these the ones that you used in developing your regression models? You did not elaborate which descriptors are used for model development (see also lines 145-150). You can possibly list all the descriptors as supplementary material.

2. The literature is not cited representatively nor discussed properly (lack of proper contextualization). Many sentences are vague and open to interpretation, with poorly defined terms particularly remote from the common hydrologic modelling terminology. Further, some arguments/discussions lack logical coherence. Here are a few examples—but not limited to this list.

a. Take the first 2 sentences of the Introduction. In the first sentence you discussed "the prediction of flow regimes in general", but in the next sentences of the paragraph you are specifically and only talking about prediction in ungauged basins. In the second sentence you mentioned "widely studied" yet you only cited 3 papers. Also, stated "prediction of hydrological data"; we do not *predict* 'data', we observe/estimate/measure them. Moreover, the methods you are referring to are only for predicting streamflow (and not other hydrological variables) in ungauged basins (and not in general).

b. Further, this very first paragraph of the Introduction, particularly lines 49-63, is filled with unsubstantiated fact-like claims. Many sentences such as lines 49-50 and 58-60 need referencing. For lines 49-50, please refer to *Ali et al.* [2012], and consider rephrasing to "hydro-climate, geomorphologic, and physical catchment characteristics (e.g., land use, soil types and geology)".

c. Line 54: what does the "process of model constitution" mean? developing a model structure, parametrization/calibration of a model?

d. Lines 55-56: term "global performance" could be interpreted in two ways: (1) performance of models on a global scale (around the globe), (2) the overall performance of model when conditioned (e.g. using global calibration methods) given a particular performance metric. Also, the term "performance parameters" is unclear to me; do you mean performance metrics that a model is evaluated against, or model (free) parameters that are calibrated for reaching a particular performance level? Moreover, it seems to me that you are referring to the problem of parameter equifinality in models here; if so, please acknowledge the terminology and its relevant literature.

e. 56-57 what "restrictive assumptions"? If the differences are very small, how they are selected then?

f. Lines 57-58: there is a rich literature on regionalization emphasizing on the importance of physical understanding of the catchment dynamics, particularly dominant runoff

generation; which you ignored here. You can refer to the extant literature discussed in *Hrachowitz et al.* [2013]. Moreover, you argued the selection criteria of the model is merely based on tradeoffs between "statistical parameters". First, what do you mean by "statistical parameters"; error metrics? Second, this is exactly what you based your model swapping on; particularly the criterion of reducing model overall error (as discussed later on table 3). Further, in many of the cases discussed in your results (presented on table 3), the improvement of swapped model ($SM_i$) over the original model (OM)—measured by root mean squared error, Nash-Sutcliffe Efficiency, and mean absolute error—is trivial, such as 28, 32 40, 44, 57, 68, 78, 82, 108, and 114. In cases that the improvement in error metric is not trivial (such as 90, 95, 101, 103, and 116), it is important to present the modelled hydrographs as well; certain aspects of the hydrograph are visually discernable, while ignored or cancelled out when numerical metrics of model evaluation are used.

g.  Lines 64-65: this seems to me like a straw man fallacy. There are many studies which used an ensemble of behavioral models and multi-objective framework to identify feasible parameterizations for ungauged basins (again, you can refer to *Hrachowitz et al.* [2013] for examples). In fact, using a single model framework is now considered as incomplete within mainstream hydrologic research/publications.

h.  Lines 101 and 145 simply do not make sense to me!

i.  Line 120 and equation 2: you stated "$L_{sp}$ which describes the time difference"; yet given the equation 2, the unit of $L_{sp}$ is not time but flow magnitude.

j.  Line 147: what do you exactly mean by "complex descriptors"? please define.

k.  For equations 5 and 6 cite the original literature [*Ganora et al.*, 2009].

l.  Lines 179: elaborate what you mean by "extraneous (junk) variable" and "standard errors".

m.  Lines 226-227: what do you mean by "better satisfies the condition of [a] meaningful transformation"?

n.  Lines 237-239, need rephrasing and substantiation, e.g. in "river flow varies unpredictably over a short distance", what do you mean by the underlined term exactly?

3.  It is not clear to me what exactly is "comprehensive" about the proposed method (mentioned multiple times e.g. lines 74, 108, 144, …). Using a FDC the "time" aspect of your hydrologic data information content is missed, yet all the observed data points are still preserved and used (i.e. their magnitude). But in the proposed method, the hydrologic data is summarized into monthly means and reduced from years of data points to a monthly so-called "representative flow regime". Arguably, you are shrinking the information content of the data in time, both in their sequencing and magnitude (by averaging).

4.  Perhaps one of the major uncertainties (or undiscussed questions) about your work is that although you tested the method on 124 basins, they are all located within a small region. That is, it is not clear how effective your proposed method would be in other regions of the world with a different hydro-climatology. Even if all the results are accurate and model predictability is improved, it is still difficult to claim that it is a superior method. So, the result discussion should be realistic, rather than over-promising about the proposed methods.

5. Equation 6: I'd like to point out to fundamental limitations of $R^2$ in assessing the predictive power (or goodness-of-fit) of regression models. First, $R^2$ is independent of model bias [*Legates and McCabe*, 1999]. Secondly, $R^2$ monotonically increases with the number of variables included in the regression (i.e. the number of descriptors $p$). In other words, $R^2$ will never decrease when you add on new descriptors to the model the number of $p$. It is useful to discuss this problem of over-fitting, and to what extent it is (ir)relevant to the methodology you proposed. Additionally, Akaike-based criteria are also useful for ranking models (from best to worst); you can see *Saft et al.* [2016] as an example.

   a. Line 163, n is not the total number of basins, it is the number of basins considered. Likewise, for equation 7 properly define n.

**Minor comments and suggestions**

1. Poor use of English language
   a. Line 54: use ; instead of :.
   b. Lines 69-71, rewrite please.
   c. Line 84: "is" tested in Italy.
   d. Stick to American or British English across the manuscript. For instance, "viz." is a British adverb in a manuscript filled with the American verb-forming suffix "-ize".
   e. Poor punctuations, e.g. using underscore '__' instead of em dash '—', e.g. lines 39, 74, 108.
2. Figure 3, location of Italy on the globe is not necessary.
3. Line 146: basin "mean" elevation
4. The term "normalized" is used throughout the literature (particularly line 140). In statistics normalization mainly refers to transforming data distribution into a normal distribution. Given the statistical nature of your work, either use a different term (e.g. transformation, re-expression, etc.), or clearly emphasize that you are not normalizing data in classical sense of the term within statistics literature. Further, d within the in-text equation should be defined.

**References**

Ali, G., D. Tetzlaff, C. Soulsby, J. J. McDonnell, and R. Capell (2012), A comparison of similarity indices for catchment classification using a cross-regional dataset, *Advances in Water Resources*, *40*, 11-22.

Ganora, D., P. Claps, F. Laio, and A. Viglione (2009), An approach to estimate nonparametric flow duration curves in ungauged basins, *Water Resources Research*, *45*(10).

Hrachowitz, M., et al. (2013), A decade of Predictions in Ungauged Basins (PUB)—a review, *Hydrological Sciences Journal*, *58*(6), 1198-1255.

Legates, D. R., and G. J. McCabe (1999), Evaluating the use of "goodness-of-fit" Measures in hydrologic and hydroclimatic model validation, *Water Resources Research*, *35*(1), 233-241.

Saft, M., M. C. Peel, A. W. Western, and L. Zhang (2016), Predicting shifts in rainfall-runoff partitioning during multiyear drought: Roles of dry period and catchment characteristics, *Water Resources Research*, *52*(12), 9290-9305.

---

## Referee Comment (RC2) · Anonymous Referee #2 · 10 Oct 2018

**Title:** A novel technique to improve the hydrological estimates at ungauged basins by swapping workspaces
**Authors:** M. U. Qamar, M. Azmat, M. Usman, D. Ganora, M. A. Shahid, F. Baig, S. Mushtaq

**OVERALL EVALUATION**

The manuscript focuses on regional streamflow regimes predictions in ungauged sites by using a dissimilarity-based method. Looking in a comprehensive way at the whole study, I regretfully have to inform the Authors that, in my opinion, the manuscript is unsuitable for publication in Hydrology and Earth System Science. My main concerns about the manuscript are listed below. I hope the Authors will find them useful should they decide in the near future to critically revise their study.

*General comment:*
Given the topic and the analyses reported, I think it is misleading to classify this manuscript as "technical note". Although the Authors do not introduce any novel method, they apply hydrological tools and models through a somewhat novel procedure, which I suppose it can ascribed to a scientific paper, rather than a technical note. I see the technical note more like a document that reports further tests on a well-known procedure or method, with the final aim to be readily available for operational purposes. Instead, I think that the procedure presented might have a potential for predicting streamflow regimes in ungauged sites, as it relies on previous studies on dissimilarity-based techniques, published by one of the co-authors (Ganora et al., 2009). Nevertheless, this potential must be further investigated. For this reason, I think it is worth the effort to work on a better in-depth comparative study, with detailed results and comparisons with other models.

*Methodological comments*:
- Even though I have understood the idea behind the whole study, I really struggled with how the swapping applies in ungauged sites. In addition, the Authors use leave-one-out cross-validation strategy (see L 170-171 P5), however they never emphasize this. In my view, given the operational purposes of the study, this should be reported better, e.g. added in the abstract and clearly stated within the body of the text.
- I am not sure about the normalization applied to either the descriptors or the discharges (i.e. hydrological variables). Variables must be comparable from one site to another. This assumption is fundamental in regional analyses, in fact, in many cases reported in the literature (among all, refer to see Blöschl et al., 2013) some sort of standardization is always employed to the streamflows, e.g. using the mean annual flow (or monthly in this case) as reference values. I might be wrong, but I do not see this step in the manuscript except for a general statement at L89-90 P3, however other normalizations seem to be used the dissimilarity indices (see L140 P4), but none for the descriptors, correct?
- I have found the mathematical notation really poor and misleading throughout the text, with some relevant inconsistencies (see e.g. how the subscripts in eq. 4 do not match with the definition in section 2).

*Other comments*:

- The manuscript is not well structured and the writing is confusing, with many errors that sometimes make very difficult the understanding of the analysis.
- Please, use English for reporting variable name (descriptors' names, see table 1). This is a minimum requirement for any manuscript, whether it is a technical note or a scientific paper.
- Since the Authors are using leave-on-out cross-validation it is possible to draw scatterplot of empirical vs. predicted dissimilarity-indices, I strongly recommend using graphical tools rather than long tables (which do not really help understanding the results), or, alternatively, please prefer summary tables in the body of text with supplementary material for the complete reporting of the results.

**REFERENCES**

[1] Blöschl, G., Sivapalan, M., Thorsten, W., Viglione, A., Savenije, H., 2013. Runoff prediction in ungauged basins: synthesis across processes, places and scales. Cambridge University Press.

[2] Ganora, D., Claps, P., Laio, F., Viglione, A., 2009. An approach to estimate nonparametric flow duration curves in ungauged   basins. Water Resour. Res. 45. https://doi.org/10.1029/2008WR007472.

---

## Short Comment (SC1) · 10 Oct 2018

We want to extend a quick gratitude to the reviewer for highly constructive criticism. We needed such a detailed review for the improvement of our work. We also thank him for providing us the option of re-submission. We assure the reviewer that his criticism is very well taken and will be addressed to the extent of his satisfaction in our revised draft. For the time being, we'll wait for the comments of the second reviewer and will inculcate all the suggest comments, simultaneously.

---

## Short Comment (SC2) · 11 Oct 2018

Dear Reviewer, Thank-you so very much for your kind comments and giving us an opportunity to resubmit our work. We will soon reply to your concerns in step-by-step manner.
* * *

---

## Author Comment (AC1) · 28 Nov 2018

Journal: Hydrology and Earth System Sciences (HESS)

Title: Technical note: A novel technique to improve the hydrological estimates at ungauged basins by swapping workspaces

MS No.: hess-2018-418 Article

Iteration: First review

**Note: The reviewer's comments are written in "blue" followed by our response in "black".**

Summary and Overall assessment

The study aims to propose a new statistical technique to improve the predictive power of multiple linear regression models used for streamflow predictions in ungauged basins. The contribution of the technique, in essence, is based on defining new (dis)similarity indices—called point-to-point, lateral, and vertical distance measures—applied to monthly flow regimes. They have used monthly flow regimes, instead of using flow duration curves (FDCs), to compare the ungauged basin with its neighboring basins arguing that FDCs loss the time aspect of hydrologic data information content. Further, they seek to improve the flow estimation in the ungauged by "swapping" the original model (used for flow estimation) by other models if certain conditions are met; namely (1) increasing the covered space around the ungauged basin, and (2) reducing the overall model error. Authors developed their regression models based on climatic and geomorphologic descriptors of neighboring basins, and applied their framework across 124 basins in northwestern Italy.

In my evaluation the manuscript is rejected, with the kind encouragement of submitting a substantially revised version. The fundamental requirement of reporting a scientific research is not met, i.e. its clarity and testability/reproducibility. The wording and phrasing on many instances within the manuscript are short of a scientific text; needless to mention that the manuscript is not even proof-read. The manuscript is filled with unsubstantiated, poorly defined, and vague statements—instances are discussed within the section below. Parts of the discussions lack logical coherence and internal consistency (comments 2.f and 2.g). The work is not properly contextualized within the extant literature. The descriptors used in developing original models are not explained properly, and the results discussion lack a discussion on the uncertainties associated with the data used, developed method, as well as the discussion of results. I encourage the authors to ask one or few experts in regionalization as well as a native English speaker to review their manuscript thoroughly before the re-submission.

Reply: We, first of all, want to thank the Reviewer for reviewing our work and then giving constructive criticism and suggestions which will lead to the improved version of our paper. We still feel that **rejection of our manuscript** based on that criticism is bit too much. However, we agree with the assessment of the reviewer and thank him for letting us submit an extensively

revised version of our manuscript. Here we want to give an emphasis on the fact that a lot of structural shortfalls which the reviewers observed are the result of our work submitted as a "Technical note". We intended to be concise and wanted to cite the most relevant literature. However, if reviewers feel that we need to elaborate more, we will do it in the revised draft while addressing all the raised concerns.

1. The manuscript is poorly structured.

Reply: We agree this general comment and our response to each individual sub-concern is below;

a) Parts of the Introduction (lines 64-82) are in fact methodology. Within the Introduction is important to discuss "why" your work matters and "why" the method is a contribution. That said, "how" the method works, should be discussed in the Methods section.
Reply: We agree with the reviewer. Lines 64-82 will be shifted to the Introduction section in the revised draft. Secondly, we believe that the method is a contribution because it allows researchers to select most appropriate model for the prediction of hydrological data through statistical and graphical signatures. This ability is unique and no other approach gives that option. In revised draft we will elaborate the stated fact more clearly.

b) The objective of the work is not clearly stated in the Introduction, instead mentioned at the end of the Study Area (lines 103-105). Further, you mentioned your primary focus is "accurate prediction". You need to exactly define what you mean by accuracy. If by accuracy you mean reducing bias, then you need to present the values of the models' biases to demonstrate the improvement (e.g. on table 3).
Reply: We completely agree with the reviewer. Lines 103-105 will be shifted to Introduction. The term accuracy is used here in a general sense i.e., bias reduction and improving statistical performance parameters (Root mean square error, Mean absolute error and Nash-Sutcliffe Efficiency). We will report biased values, graphically, in the revised draft.

c) The manuscript does not have a Methods section. To better elaborate and emphasize your proposed method, I think it is essential to put sections 3-5 under an overarching section of Methods, consolidate them (some parts are hard to follow) and remove repetitions, and perhaps before section 2 Study Area. Further, the procedure explained within lines 21-221 (including the figure) is hard to follow. Please rephrase and explain more clearly.
Reply: As we mentioned before that our main objective was not to submit a full fledge paper having general contents but rather a technical note explaining the utility of a novel technique. The criticism related to the structure of the paper is the result of our preference to submit it as a technical note. However, on the

advice of the reviewer we will put the sections 3-5 under methodology followed by consolidating it, in the revised draft. Moreover, the Lines 21-221 will be explained more clearly and in relatively easy terms/ language.

d) Lines 93-95: you listed "some of the descriptors (out of 74 descriptors)" on table 1. Why did you choose these descriptors? Are these the ones that you used in developing your regression models? You did not elaborate which descriptors are used for model development (see also lines 145-150). You can possibly list all the descriptors as supplementary material.

We agree with the latter part of the comment. The list of descriptors used in our work will be provided in the revised draft. The descriptors we provided in Table 1 are all those which are used in our models. The comprehensive compilation of geomorphological and climatic descriptors obtained for all the selected basins of the study area is provided in the literature (Gallo et al., 2013; Farr et al., 2007).

**References:**

Gallo, E., Ganora, D., Laio, F., Masoero, A., and Claps, P.: Atlante dei bacini imbriferi piemontesi (Atlas of river basins in Piemonte) Regione Piemonte, ISBN 978-88-96046-06-7, 2013.

Farr, T.G., et al.: The Shuttle Radar Topography Mission, Rev. Geophys., 45, RG2004, doi:10.1029/2005RG000183, 2007.

2. The literature is not cited representatively nor discussed properly (lack of proper contextualization). Many sentences are vague and open to interpretation, with poorly defined terms particularly remote from the common hydrologic modelling terminology. Further, some arguments/discussions lack logical coherence. Here are a few examples—but not limited to this list.

We will get rid of vague sentences in the revised draft and will establish a logical coherence too.

a. Take the first 2 sentences of the Introduction. In the first sentence, you discussed "the prediction of flow regimes in general", but in the next sentences of the paragraph you are specifically and only talking about prediction in ungauged basins. In the second sentence, you mentioned "widely studied" yet you only cited 3 papers. Also, stated "prediction of hydrological data"; we do not predict 'data', we observe/estimate/measure them. Moreover, the methods you are referring to are only for predicting streamflow (and not other hydrological variables) in ungauged basins (and not in general).

Reply: Actually, in the first sentences we referred to the general utility of the topic.

Whereas, in the next sections we elaborated topic, which is the prediction of the flow regimes.

Secondly, we cited three papers because all these papers are latest contributions in the field of predictive hydrology and use three completely different methods for the prediction of flow regimes. However, as per the directions of the reviewers we will cite more papers in the revised draft.

Thirdly, the method we are proposing is to be used for the prediction of hydrological data, whereas, the actual data is measured/ estimate/ or observed.

Finally, yes! the method is to be used for the prediction of flow regimes. We will address this concern is the revised draft.

b.  Further, this very first paragraph of the Introduction, particularly lines 49-63, is filled with unsubstantiated fact-like claims. Many sentences such as lines 49-50 and 58-60 need referencing. For lines 49-50, please refer to Ali et al. [2012], and consider rephrasing to "hydro-climate, geomorphologic, and physical catchment characteristics (e.g., land use, soil types and geology)".

    Reply: We will provide the references of the statements in the revised draft. We will also rephrase some of the terms we used in our paper.

c.  Line 54: what does the "process of model constitution" mean? developing a model structure, parametrization/calibration of a model?

    Reply: Model development means developing, calibrating, and validating of a model.

d.  Lines 55-56: term "global performance" could be interpreted in two ways: (1) performance of models on a global scale (around the globe), (2) the overall performance of model when conditioned (e.g. using global calibration methods) given a particular performance metric. Also, the term "performance parameters" is unclear to me; do you mean performance metrics that a model is evaluated against, or model (free) parameters that are calibrated for reaching a particular performance level? Moreover, it seems to me that you are referring to the problem of parameter equifinality in models here; if so, please acknowledge the terminology and its relevant literature.

    Reply: The global performance of the models mean the latter, i.e., the overall performance of model when conditioned (e.g. using global calibration methods) given a particular performance metric. The term "performance parameters" refers to the statistical performance parameters which evaluate model performance i.e., Root-mean Square Error, Mean Absolute Error and Nash–Sutcliffe model

efficiency. The relevant literature to the problem of parameter equifinality in models will be referred in the revised draft.

e. 56-57 what "restrictive assumptions"? If the differences are very small, how they are selected then?

The equations (6) and (7) execute $R^2_{adj}$ and $\Delta$ values, respectively. These values are used to identify the model with better global performance (Original Model). The model with comparatively higher $R^2_{adj}$ and least $\Delta$ value is selected to make initial estimation.

f. Lines 57-58: there is a rich literature on regionalization emphasizing on the importance of physical understanding of the catchment dynamics, particularly dominant runoffgeneration; which you ignored here. You can refer to the extant literature discussed in Hrachowitz et al. [2013]. Moreover, you argued the selection criteria of the model is merely based on tradeoffs between "statistical parameters". First, what do you mean by "statistical parameters"; error metrics? Second, this is exactly what you based your model swapping on; particularly the criterion of reducing model overall error (as discussed later on table 3). Further, in many of the cases discussed in your results (presented on table 3), the improvement of swapped model (SMi) over the original model (OM)—measured by root mean squared error, Nash-Sutcliffe Efficiency, and mean absolute error—is trivial, such as 28, 32 40, 44, 57, 68, 78, 82, 108, and 114. In cases that the improvement in error metric is not trivial (such as 90, 95, 101, 103, and 116), it is important to present the modelled hydrographs as well; certain aspects of the hydrograph are visually discernable, while ignored or cancelled out when numerical metrics of model evaluation are used.

Reply: Hrachowitz et al. [2013] will be cited in the revised draft. By "statistical parameters" we meant goodness of fit and error statistics. The model selection criteria are not strictly defined and require cross validation even after a strong goodness of fit test. We built our argument on the fact that since the criteria is not strictly defined therefore selecting one model when performances of other model are match able is over simplification. We believe that instead of selecting one model, we should select all the models having almost similar global performances and then swap them to predict hydrological data for individual basin provided the set criteria of swapping are fulfilled.

g. Lines 64-65: this seems to me like a straw man fallacy. There are many studies which used an ensemble of behavioral models and multi-objective framework to identify feasible parameterizations for ungauged basins (again, you can refer to

Hrachowitz et al. [2013] for examples). In fact, using a single model framework is now considered as incomplete within mainstream hydrologic research/publications.

Reply: We request reviewer to review this statement in the context of our work. Within the paradigm of distance-based method, using single regression model is still practiced. So much so, that recent works published in some prestigious scientific journals used nearest neighbor method in a single model framework complimented by Euclidean distance workspace (Geostatistical methods). Therefore, using single model was only referred to the distance-based methods. Some of the works are referred below;

Archfield, S.A.; Pugliese, A.; Castellarin, A.; Skøien, J.O. Topological and canonical kriging for design-flood prediction in ungauged catchments: An improvement over a traditional regional regression approach? Hydrol. Earth Syst. Sci. 2013, 9, 1575–1588.

Castiglioni, S.; Castellarin, A.; Montanari, A. Prediction of low-flow indices in ungauged basins through physiographical space-based interpolation. J. Hydrol. 2009, 378, 272–280.

h.  Lines 101 and 145 simply do not make sense to me!

Line 101: The flow regime thus obtained represents streamflow variability in each month.
Line 145: Unlike flow data, the descriptors data is varying in nature due to different types of descriptors introduced recently (e.g., geomorphological descriptors, climatic descriptors, etc.).

i.  Line 120 and equation 2: you stated "Lsp which describes the time difference"; yet given the equation 2, the unit of Lsp is not time but flow magnitude.
The Lsp represents the lateral separation of the peaks. To estimate the lateral separation, the regimes were moved toward each-other until the peaks are exactly underneath and there is no lateral separation. Each step of peak shifting is quantified into dissimilarity magnitude by observing the dissimilarity between the regimes before and after the shifting. Since Lsp is a form of dissimilarity therefore it is read from the y-axis. In other words, the time difference in the occurrence of peaks is converted into dissimilarity to execute comprehensive dissimilarity between flow regimes.

j.  Line 147: what do you exactly mean by "complex descriptors"? please define.

Complex descriptors are those descriptors for which dissimilarity cannot be executed by using simple comparison (rainfall regime, hypsographic curve) and require some other methods to execute the dissimilarity (equations 6 or 7).

k. For equations 5 and 6 cite the original literature [Ganora et al., 2009].

We agree with the review. The concern will be addressed in the revised draft.

l. Lines 179: elaborate what you mean by "extraneous (junk) variable" and "standard errors".

The 2D workspace is formulated by two descriptors. Sometimes, a number of basins have equal values of a certain descriptor. In case some of the nearest neighbors of ungauged basin have duplication in any of the descriptor values, the dissimilarity magnitude between the basins is dominated by one descriptor (so-called junk) variable constituting the predicting model resulting in inflated standard errors.

m. Lines 226-227: what do you mean by "better satisfies the condition of [a] meaningful transformation"?

The mean of hydrological data of $NNs$ of $u_g$ in the workspace of the models is always converged to the center (by definition of mean). When the same phenomenon is repeated by the actual position of $u_g$ and its NNs in the descriptors space we call it meaning transformation. Ideally, the position of $u_g$ in hydrological workspace (which is always in the middle as $u_g$ hydrological data is executed by taking the mean of its NNs) should be replicated in the descriptor workspace as well. The descriptor workspace in which $u_g$ is located closest to the center formed by the (average of) descriptor values of its NNs is considered better.

n. Lines 237-239, need rephrasing and substantiation, e.g. in "river flow varies unpredictably over a short distance", what do you mean by the underlined term exactly?

Here we argue that the hydrologic response of a basin varies unpredictably under the effect of basin characteristics (descriptor values). The unpredictable climatic and geomorphological factors including human impact, contribute to seasonal river discharge (Pavlov 1994, 1996). Whereas, "over a shorter distance" is written by mistake and will be eliminated in the revised version.

**References:**

Pavlov, A. V., 1994: Current changes of climate and permafrost in the Arctic and sub-Arctic of Russia. Permafrost Periglacial Processes, 5, 101–110.

Pavlov, A. V., 1994: Permafrost-climatic monitoring of Russia: Analysis of field data and forecast. Polar Geogr. Geol., 20, 44–64.

3. It is not clear to me what exactly is "comprehensive" about the proposed method (mentioned multiple times e.g. lines 74, 108, 144, …). Using a FDC the "time" aspect of your hydrologic data

information content is missed, yet all the observed data points are still preserved and used (i.e. their magnitude). But in the proposed method, the hydrologic data is summarized into monthly means and reduced from years of data points to a monthly so-called "representative flow regime". Arguably, you are shrinking the information content of the data in time, both in their sequencing and magnitude (by averaging).

The word comprehensively is used here in the context of dissimilarity. The method comprehensively defined the dissimilarity between the flow regimes by comparing magnitudes of flows in flow regimes, lateral separation of peaks and vertical separation of peaks. As far as reducing 365 flow values to 12 values is concerned, it totally depends on the objectives of a certain methodology. Our objectives were to predict the flow magnitudes in regimes and then to temporally estimate the peak flow.

4. Perhaps one of the major uncertainties (or undiscussed questions) about your work is that although you tested the method on 124 basins, they are all located within a small region. That is, it is not clear how effective your proposed method would be in other regions of the world with a different hydro-climatology. Even if all the results are accurate and model predictability is improved, it is still difficult to claim that it is a superior method. So, the result discussion should be realistic, rather than over-promising about the proposed methods.

We completely agree with the reviewer. The methodology is only applied in the northwestern part of the Italy. Although even in the present draft we have summed up the limitations of our methodology, in the revised draft we will include a whole sub-section in the "Discussion section" on the shortcomings of our work for the researchers to address them in the future.

5. Equation 6: I'd like to point out to fundamental limitations of R2 in assessing the predictive power (or goodness-of-fit) of regression models. First, R 2 is independent of model bias [Legates and McCabe, 1999]. Secondly, R 2 monotonically increases with the number of variables included in the regression (i.e. the number of descriptors p). In other words, R 2 will never decrease when you add on new descriptors to the model the number of p. It is useful to discuss this problem of over-fitting, and to what extent it is (ir)relevant to the methodology you proposed. Additionally, Akaike-based criteria are also useful for ranking models (from best to worst); you can see Saft et al. [2016] as an example.

We completely agree with the reviewer. Owing to the above mentioned criticism on $R^2$, the model selection criterion (based on $R^2$ values of the models) is backed up with the ($\Delta$) factor, which indicates the mean error magnitude produced by the model for the entire study area excluding the ungauged basin. The aim of this whole exercise is to reduce the error between actual and predicted flow regimes at ungauged sites. This said, the reviewer has done pretty useful criticism on $R^2$ and we will include his comment in the revised draft.

a. Line 163, n is not the total number of basins, it is the number of basins considered. Likewise, for equation 7 properly define n.

It is pertinent to mention that (Ganora et al., 2009) used the same equation and defined "n" as number of basins considered as pointed out the reviewer. (Ganora et al., 2009) divided the workspace into clusters and analysis was carried out in each cluster. That's why they defined "n" as the number of basins considered. However, in our work we considered all the basin for our analysis unlike (Ganora et al., 2009). Therefore, in our work "n" truly is the total number of basins.

Minor comments and suggestions

1. Poor use of English language

a. Line 54: use ; instead of :.

Agree and this will be addressed in the revised draft.

b. Lines 69-71, rewrite please.

Lines 69-71 define the generic difference between flow duration curves and fow regimes. They will be rephrased in the revised draft.

c. Line 84: "is" tested in Italy.

Agree and this will be addressed in the revised draft.

d. Stick to American or British English across the manuscript. For instance, "viz." is a British adverb in a manuscript filled with the American verb-forming suffix "-ize".

Agree and this will be addressed in the revised draft.

e. Poor punctuations, e.g. using underscore '__' instead of em dash '—', e.g. lines 39, 74, 108.

Agree and this will be addressed in the revised draft.

2. Figure 3, location of Italy on the globe is not necessary.

Agree and this will be addressed in the revised draft.

3. Line 146: basin "mean" elevation

Agree and this will be addressed in the revised draft.

4. The term "normalized" is used throughout the literature (particularly line 140). In statistics normalization mainly refers to transforming data distribution into a normal distribution. Given the statistical nature of your work, either use a different term (e.g. transformation, re-expression, etc.), or clearly emphasize that you are not normalizing data in classical sense of the term within statistics literature. Further, d within the in-text equation should be defined.

We agree with the reviewer. We will use the term "Standardization" in our revised draft. Moreover, d within the text stands for a "descriptor".

---

## Author Comment (AC2) · 28 Nov 2018

Reviewer # 2

Title: A novel technique to improve the hydrological estimates at ungauged basins by swapping workspaces

Authors: M. U. Qamar, M. Azmat, M. Usman, D. Ganora, M. A. Shahid, F. Baig, S. Mushtaq

**Note: The reviewer's comments are written in "blue" followed by our response in "black".**

OVERALL EVALUATION

The manuscript focuses on regional streamflow regimes predictions in ungauged sites by using a dissimilarity-based method. Looking in a comprehensive way at the whole study, I regretfully have to inform the Authors that, in my opinion, the manuscript is unsuitable for publication in Hydrology and Earth System Science. My main concerns about the manuscript are listed below. I hope the Authors will find them useful should they decide in the near future to critically revise their study.

General comment: Given the topic and the analyses reported, I think it is misleading to classify this manuscript as "technical note". Although the Authors do not introduce any novel method, they apply hydrological tools and models through a somewhat novel procedure, which I suppose it can ascribed to a scientific paper, rather than a technical note. I see the technical note more like a document that reports further tests on a well-known procedure or method, with the final aim to be readily available for operational purposes. Instead, I think that the procedure presented might have a potential for predicting streamflow regimes in ungauged sites, as it relies on previous studies on dissimilarity-based techniques, published by one of the co-authors (Ganora et al., 2009). Nevertheless, this potential must be further investigated. For this reason, I think it is worth the effort to work on a better in-depth comparative study, with detailed results and comparisons with other models.

Reply: First of all, we thank reviewer for positive criticism. By and large, we agree with the reviewer. However, as the worthy reviewer himself/ herself discussed in the later part of the comment that the procedure relies on previous studies done in the context of dissimilarity-based hydrological prediction, published by one of the co-authors (Ganora et al., 2009). For this sole reason, we opted to go for submission as a technical note instead of a full fledge paper. The suggestion of reviewer about the comparison of our method with other established procedures is very useful. In the revised draft, we will briefly compare the performance of our method with other practiced methods.

Methodological comments:

Even though I have understood the idea behind the whole study, I really struggled with how the• swapping applies in ungauged sites. In addition, the Authors use leave-one-out cross-validation strategy (see L 170-171 P5), however they never emphasize this. In my view, given the

operational purposes of the study, this should be reported better, e.g. added in the abstract and clearly stated within the body of the text.

Reply: We agree with the reviewer. We elaborate a step-wise procedure for the application of the proposed methodology on the ungauged basins;

1) Selection of Original and Swapped models by considering $\Delta$ and $R^2_{adj}$.

2) Estimating $\Delta_{NN}$ and $C_f$ values for each basin using OM and SM. Where, $\Delta_{NN}$ represents the error generated in predicting hydrological data of NNs of $u_g$ in the cluster formed by $u_g^{NN}$ and NNs of $u_g^{NN}$ (using hydrological data although unknow for $u_g$ but known for its NNs ($u_g^{NN}$) as well as NNs of $u_g^{NN}$). Whereas, $C_f$ represents the coverage factor created by NNs of $u_g$ in the descriptor space (using descriptor data know for NNs as well as $u_g$).

3) Prefer SM over OM iff $\Delta_{NN}^{OM} > \Delta_{NN}^{SM}$; and $C_f^{OM} < C_f^{SM}$.

Although these points are already inculcated in the text but we will elaborate these points more comprehensively in the revised draft.

I am not sure about the normalization applied to either the descriptors or the discharges (i.e. hydrological variables). Variables must be comparable from one site to another. This assumption is fundamental in regional analyses, in fact, in many cases reported in the literature (among all, refer to see Blöschl et al., 2013) some sort of standardization is always employed to the streamflows, e.g. using the mean annual flow (or monthly in this case) as reference values. I might be wrong, but I do not see this step in the manuscript except for a general statement at L89-90 P3, however other normalizations seem to be used the dissimilarity indices (see L140 P4), but none for the descriptors, correct?

Reply: We completely agree with the reviewer. Indeed, some sort of standardization is always employed to the streamflows, e.g. using the mean annual flow (or monthly in this case) as reference values to make variables must be comparable from one site to another. The descriptors and hydrological data in our work are normalized by using mean values at each basin.

I have found the mathematical notation really poor and misleading throughout the text, with some relevant inconsistencies (see e.g. how the subscripts in eq. 4 do not match with the definition in section 2).

Reply: This comment has also been raised by the reviewer 1. In revised draft, we will make sure to remove these inconsistencies.

Other comments:

The manuscript is not well structured and the writing is confusing, with many errors that sometimes make very difficult the understanding of the analysis.

Reply: We completely agree with the reviewer. Honestly speaking, we tried to make the article as short as possible to make it look like a technical note. However, with serious concerns raised over the structure we are compelled to restructure it. The revised draft will be restructured carrying more general contents.

Please, use English for reporting variable name (descriptors' names, see table 1). This is a minimum requirement for any manuscript, whether it is a technical note or a scientific paper.

Reply: The revised draft will have all descriptors notated in English.

Since the Authors are using leave-on-out cross-validation it is possible to draw scatterplot of empirical vs. predicted dissimilarity-indices, I strongly recommend using graphical tools rather than long tables (which do not really help understanding the results), or, alternatively, please prefer summary tables in the body of text with supplementary material for the complete reporting of the results.

Reply: We agree with the reviewer. We will add summary tables and scatterplots in the revised draft. We initially thought of adding the scatterplots. However, we ditched the idea as this will considerably increase the length of the manuscript owing to the fact that we will have to add multiple graphs between different variables.